# Pre-training LLM without Learning Rate Decay Enhances Supervised Fine-Tuning

**Kazuki Yano**[†]**, Shun Kiyono**[‡]**, Sosuke Kobayashi**[†]**, Sho Takase**[†]**, Jun Suzuki**[†]
[†]Tohoku University
[‡]SB Intuitions
`yano.kazuki@dc.tohoku.ac.jp`  `is-failab-research@grp.tohoku.ac.jp`

## Abstract

We investigate the role of learning rate scheduling in the large-scale pre-training of large language models, focusing on its influence on downstream performance after supervised fine-tuning (SFT). Decay-based learning rate schedulers are widely used to minimize pre-training loss. However, despite their widespread use, how these schedulers affect performance after SFT remains underexplored. In this paper, we examine Warmup-Stable-Only (WSO), which maintains a constant learning rate after warmup without any decay. Through experiments with 1B and 8B parameter models, we show that WSO consistently outperforms decay-based schedulers in terms of performance after SFT, even though decay-based schedulers may exhibit better performance after pre-training. The result also holds across different regimes with mid-training and over-training. Loss landscape analysis further reveals that decay-based schedulers lead models into sharper minima, whereas WSO preserves flatter minima that support adaptability. These findings indicate that applying LR decay to improve pre-training metrics may compromise downstream adaptability. Our work also provides practical guidance for training and model release strategies, highlighting that pre-training models with WSO enhances their adaptability for downstream tasks.

## 1 Introduction

Learning rate (LR) scheduling is arguably one of the most critical yet operationally challenging aspects of large language model (LLM) pre-training. Although Cosine decay has been conventionally employed in numerous models (Brown et al., 2020; Le et al., 2022; Touvron et al., 2023a), it has proven inflexible in recent training paradigms such as continual pre-training, as it requires heuristic tuning of the LR from the decayed value (Hägele et al., 2024; Ibrahim et al., 2024). To address this inflexibility, recent studies have introduced Warmup-Stable-Decay (WSD), which keeps the LR constant through most of pre-training and decays it only briefly at the end (Hu et al., 2024; Liu et al., 2024a; Wen et al., 2025).

These previous studies, regardless of the details of the design choices, decayed the LRs to optimize the performance of pre-trained models. However, the more critical factor for real applications is the performance after post-training, such as supervised fine-tuning (SFT). Drawing on the findings of Sun & Dredze (2025) and Springer et al. (2025), which show that a strong pre-training model does not necessarily imply superior performance after SFT, it is questionable to schedule LRs to the decayed value based on pre-training performance.

In this study, we provide a comprehensive empirical investigation of LR schedulers during pre-training in terms of performance after SFT. In particular, we examine an underestimated scheduling, Warmup-Stable-Only (WSO), which removes the decay phase from WSD and maintains constant LR to the end. We show that WSO consistently achieves superior performance after SFT compared to decay-based schedulers, through experiments on 1B and 8B models (Figure 1). Furthermore, we demonstrate that WSO is also effective under modern training paradigms, including mid-training (OLMo et al., 2024; Meta, 2024c) and over-training (Sardana et al., 2024; Gadre et al., 2025).

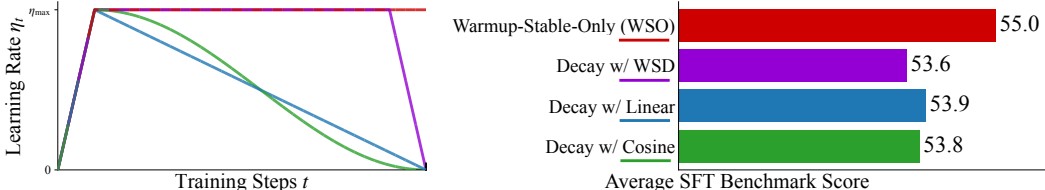

Figure 1: Learning rate schedulers used in pre-training and their impact on performance after supervised fine-tuning (SFT). Warmup-Stable-Only (WSO), which removes the decay phase, achieves the highest performance after SFT.

To understand why WSO yields superior SFT performance, we draw on insights from the transfer learning literature (Ju et al., 2022; Liu et al., 2023), which suggest that models in flatter regions of the loss landscape tend to exhibit better adaptability. Through an analysis of sharpness values, we show that models trained with WSO reside in flatter regions than those trained with other decay-based LR schedulers, and are therefore more adaptable to post-training tasks.

Our contributions are as follows: (1) We provide the systematic demonstration that WSO consistently outperforms decay-based schedulers on downstream tasks after SFT, with comprehensive evidence across 1B and 8B models and diverse evaluation benchmarks. (2) We show that WSO similarly benefits mid-training and over-training scenarios, achieving superior SFT performance compared to conventional decay-based schedulers. (3) We reveal through loss landscape analysis that WSO preserves flatter minima than decay-based schedulers, explaining why models trained with WSO achieve better performance after SFT.

## 2 PRELIMINARIES

Recent LLMs are typically built with a staged training scheme. The most common and fundamental training pipeline consists of two stages, namely pre-training and post-training. In this section, we describe these training stages and review the LR schedulers commonly employed during pre-training.

**Pre-training.** Pre-training forms the foundation of LLM development, where models learn general language understanding from massive text corpora by minimizing the next-token prediction loss. Recently, pre-training has sometimes consisted of multiple stages: standard pre-training and mid-training (OLMo et al., 2024). We describe mid-training in detail later (Section 2.2), and conduct experiments with both the standard pre-training and the multi-stage setup.

**Post-training.** Post-training adapts pre-trained models to target tasks, enabling them to follow human instructions and avoid generating harmful outputs. Post-training includes techniques such as supervised fine-tuning (SFT), preference tuning (e.g., DPO (Rafailov et al., 2023)), and RL-based alignment (Ouyang et al., 2022). While post-training could be a multi-stage process with many design choices still under active exploration, SFT is relatively standardized and serves a core stage. In this paper, we focus on SFT as the canonical post-training stage and evaluate the performance after SFT[1].

### 2.1 TASK DEFINITION

Practically, LLM developers evaluate models at multiple stages, selecting the best-performing one as the starting point for the subsequent stage. We define $\text{Task}_s(M)$ as a function that, for a given LLM $M$, returns the performance on a set of pre-defined tasks used to assess the target stage $s$, where

---

[1]The computational cost of pre-training is typically much larger than that of other stages, so identifying a better pre-training configuration has a substantial impact on the efficiency of LLM construction. In this study, we focus on evaluating LR schedulers during large-scale pre-training and characterize the potential of non-decay schedulers based on the performance after SFT. An exploration of complex combinations of LR scheduling spanning multiple post-training stages is left to future work.

$s \in \{\text{pre}, \text{post}\}$ denotes the training stage, with $\text{pre}$ indicating pre-training and $\text{post}$ indicating post-training, respectively. We write $M_2[M_1]$ to denote the model $M_2$ trained with some configuration and initialization with $M_1$, where $M_{\text{rand}}$ indicates a model whose weights are randomly initialized. Moreover, we introduce $\mathcal{M}_{\text{pre}}$ and $\mathcal{M}_{\text{post}}$ to represent the sets of models obtained through pre-training and post-training, respectively, with various hyperparameter configurations. A typical training pipeline for building LLMs can therefore be expressed as follows:

$$
\begin{aligned}
\widehat{M}_{\text{pre}} &= \underset{M_{\text{pre}} \in \mathcal{M}_{\text{pre}}}{\arg\max} \left\{ \texttt{Task}_{\text{pre}}(M_{\text{pre}}[M_{\text{rand}}]) \right\}, \\
\widehat{M}_{\text{post}} &= \underset{M_{\text{post}} \in \mathcal{M}_{\text{post}}}{\arg\max} \left\{ \texttt{Task}_{\text{post}}(M_{\text{post}}[\widehat{M}_{\text{pre}}[M_{\text{rand}}]]) \right\}.
\end{aligned}
\tag{1}
$$

This formulation may lead to a suboptimal solution in terms of the performance of the final model, namely, $\widehat{M}_{\text{post}}$, since selecting the best-performing models at intermediate stages does not guarantee achieving the best performance in the end. Therefore, conceptually, we would like to consider the following search problem to obtain a better final model for this training pipeline:

$$
\widehat{M}_{\text{post}} = \underset{(M_{\text{pre}}, M_{\text{post}}) \in (\mathcal{M}_{\text{pre}}, \mathcal{M}_{\text{post}})}{\arg\max} \left\{ \texttt{Task}_{\text{post}}(M_{\text{post}}[M_{\text{pre}}[M_{\text{rand}}]]) \right\}.
\tag{2}
$$

The primary objective of this paper is to empirically examine the search problem by evaluating several LR schedulers during the large-scale training stages that precede post-training.

## 2.2 FURTHER CONSIDERATIONS

**Mid-training.** Mid-training has emerged as a critical intermediate stage in modern language model development, occupying a computational middle ground between large-scale pre-training and task-specific post-training (Meta, 2024c; OLMo et al., 2024). This stage serves multiple strategic objectives, including domain expansion and long-context extension. For example, OLMo 2 (OLMo et al., 2024) demonstrates performance gains through mid-training on curated high-quality data, establishing this stage as an essential component of the modern training pipeline. After introducing mid-training, we can rewrite equation 2 as follows:

$$
\widehat{M}_{\text{post}} = \underset{(M_{\text{pre}}, M_{\text{mid}}, M_{\text{post}}) \in (\mathcal{M}_{\text{pre}}, \mathcal{M}_{\text{mid}}, \mathcal{M}_{\text{post}})}{\arg\max} \left\{ \texttt{Task}_{\text{post}}(M_{\text{post}}[M_{\text{mid}}[M_{\text{pre}}[M_{\text{rand}}]]]) \right\}.
\tag{3}
$$

**Over-training.** Modern LLMs are often trained on trillions of tokens, far beyond the Chinchilla compute-optimal regime of roughly 20 tokens per parameter (Hoffmann et al., 2022). This practice trades substantially more training compute for improved inference efficiency at deployment. Recent production systems use hundreds to thousands of tokens per parameter (Sardana et al., 2024). While full-scale experiments are costly, Section 5 presents results under such a configuration, showing the generality of our main findings.

## 2.3 CURRENT LR SCHEDULING PRACTICES

In current LLM training practice, pre-training uses decay-based LR schedulers with Cosine, Linear, or WSD that reduce LR to 0–10% of maximum (Touvron et al., 2023a; Hu et al., 2024; Bergsma et al., 2025). Additionally, in mid-training, it is common practice to further decay the LR from the final value reached at the end of the preceding pre-training phase (Meta, 2024c; OLMo et al., 2024). These schedulers are chosen to minimize loss at each respective stage, effectively optimizing $\texttt{Task}_{\text{pre}}(M_{\text{pre}})$ independently. However, the primary objective should be to maximize $\texttt{Task}_{\text{post}}(M_{\text{post}})$, the performance after the complete pipeline. Thus, optimizing for $\texttt{Task}_{\text{pre}}(M_{\text{pre}})$ may be suboptimal. For instance, recent findings from Springer et al. (2025) and Sun & Dredze (2025) reveal that the better performance after pre-training does not guarantee performance after SFT. These raise a fundamental question: *Is LR decay, which is chosen based on pre-training performance, still the best choice when the model will undergo supervised fine-tuning?* Our work investigates this question by systematically varying LR schedulers in $\mathcal{M}_{\text{pre}}$ and $\mathcal{M}_{\text{mid}}$ to understand their impact on the final objective, i.e., $\texttt{Task}_{\text{post}}(M_{\text{post}})$.

## 2.4 FORMALIZATION OF LEARNING RATE SCHEDULERS

We denote the LR at training step $t$ as $\eta^{\texttt{Scheduler}}(t, \alpha_{\text{pre}})$, where $\texttt{Scheduler}$ specifies the LR scheduler and $\alpha_{\text{pre}}$ controls the minimum LR factor in pre-training. For example, the WSD scheduler is defined as:

$$\eta^{\texttt{WSD}}(t, \alpha_{\text{pre}}) = \begin{cases} \eta_{\max} \cdot \frac{t}{T_{\text{warmup}}} & t \leq T_{\text{warmup}} \\ \eta_{\max} & T_{\text{warmup}} < t \leq T_{\text{stable}} \\ \eta_{\max} \cdot \left( (1 - \alpha_{\text{pre}}) \cdot \frac{T_{\text{pre}} - t}{T_{\text{pre}} - T_{\text{stable}}} + \alpha_{\text{pre}} \right) & T_{\text{stable}} < t \leq T_{\text{pre}} \end{cases} \quad (4)$$

where $\eta_{\max}$ is the maximum LR, $T_{\text{pre}}$ denotes the total number of pre-training steps, $T_{\text{warmup}}$ is the number of warmup steps, and $T_{\text{stable}}$ is the step at which the decay phase begins.

To investigate the effectiveness of the LR scheduler without decay, we consider a simple variant of WSD, which we call Warmup-Stable-Only (WSO). In this variant, the decay phase is omitted, which corresponds to setting $\alpha_{\text{pre}} = 1.0$.

$$\eta^{\texttt{WSO}}(t, \alpha_{\text{pre}}) = \begin{cases} \eta_{\max} \cdot \frac{t}{T_{\text{warmup}}} & t \leq T_{\text{warmup}} \\ \eta_{\max} & T_{\text{warmup}} < t \leq T_{\text{pre}} \end{cases} \quad (5)$$

In our experiments, we investigate four LR schedulers: $\texttt{Scheduler} \in \{\text{WSO}, \text{WSD}, \text{Cosine}, \text{Linear}\}$. The detailed formulations for Cosine $\eta^{\texttt{Cosine}}(t, \alpha_{\text{pre}})$ and Linear $\eta^{\texttt{Linear}}(t, \alpha_{\text{pre}})$ are provided in Appendix B.

**LR Scheduling in Mid-training.** We parameterize mid-training schedulers with $\alpha_{\text{mid}}$ (Figure 2), where $\alpha_{\text{mid}} = 0.0$ applies Linear decay to zero while $\alpha_{\text{mid}} = 1.0$ maintains the LR constant throughout mid-training. When combined with $\alpha_{\text{pre}} = 1.0$, the configuration of $\alpha_{\text{pre}} = 1.0$ and $\alpha_{\text{mid}} = 1.0$ extends WSO across both pre-training and mid-training stages. The detailed formulation for mid-training LR schedulers is provided in Appendix B.

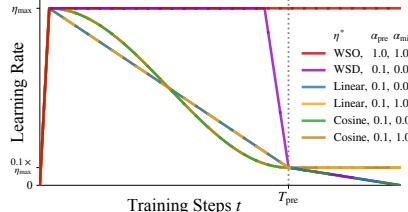

Figure 2: Mid-training LR schedulers with different $\alpha_{\text{pre}}$ and $\alpha_{\text{mid}}$ values.

## 3 EXPERIMENT 1: TWO-STAGE (PRE- AND POST-TRAINING) SETTING

We investigate whether decaying LRs during pre-training truly benefit downstream SFT performance.

### 3.1 EXPERIMENTAL SETUP

**Model Architectures.** We conduct experiments on two model scales following the Llama 3 architecture family: 1B and 8B parameter models (same architecture as Llama-3.2-1B (Meta, 2024b) and Llama-3.1-8B (Meta, 2024a), respectively). Full details are provided in Appendix A.

**Pre-training Configuration.** Models are pre-trained on FineWeb-Edu (Penedo et al., 2024) with a maximum LR $\eta_{\max} = 3 \times 10^{-4}$. We investigate three LR schedulers as formalized in Section 2.4, experimenting with WSO (Equation 2.4), WSD (Equation 2.4), Cosine, and Linear schedulers (detailed in Appendix B). For each scheduler, we vary the minimum LR factor $\alpha_{\text{pre}} \in \{0.0, 0.1, 1.0\}$, following our notation $\eta^{\texttt{Scheduler}}(t, \alpha_{\text{pre}})$. Setting $\alpha_{\text{pre}} = 0.0$ corresponds to decay to zero. Recent work by Bergsma et al. (2025) shows that this achieves better pre-training performance. Setting $\alpha_{\text{pre}} = 0.1$ corresponds to decay to 10% of maximum, a choice commonly used in practice by Chinchilla (Hoffmann et al., 2022), Llama 3 (Meta, 2024c) and OLMo 2 (OLMo et al., 2024). Finally, setting $\alpha_{\text{pre}} = 1.0$ corresponds to WSO. Further hyperparameter details are provided in Appendix C.

Table 1: Relative performance across pre-training (PT) and supervised fine-tuning (SFT). For each model size and each metric, values are differences ($\Delta$) from the best-performing decay-based scheduler for that metric. Note that WSO could perform poorly after PT *but best after SFT*. Bold indicates the best performance.

| Model | Scheduler | $\alpha_{\text{pre}}$ | PT Valid Loss $\downarrow \Delta$ | PT Task Avg $\Delta$ | SFT Task Avg $\Delta$ |
|---|---|---|---|---|---|
| | Warmup-Stable-Only (WSO) | 1.0 | *+0.071* | *-1.7* | ***+0.3*** |
| | WSD | 0.1 | +0.004 | -1.5 | +0.0 |
| | | 0.0 | **+0.000** | -1.2 | -1.0 |
| 1B | Linear | 0.1 | +0.021 | -2.0 | -0.7 |
| | | 0.0 | +0.016 | **+0.0** | -0.9 |
| | Cosine | 0.1 | +0.019 | -0.1 | -0.7 |
| | | 0.0 | +0.016 | -2.5 | -0.7 |
| | Warmup-Stable-Only (WSO) | 1.0 | *+0.127* | *-0.8* | ***+1.1*** |
| | WSD | 0.1 | +0.019 | -0.2 | -0.8 |
| | | 0.0 | +0.014 | **+0.0** | -0.3 |
| 8B | Linear | 0.1 | +0.013 | -1.9 | -0.6 |
| | | 0.0 | **+0.000** | -1.8 | +0.0 |
| | Cosine | 0.1 | +0.009 | -2.2 | -0.3 |
| | | 0.0 | +0.008 | -2.3 | -0.1 |

**SFT Configuration.** We perform SFT using the Tulu-3 SFT mixture[2]. We conduct a comprehensive LR sweep ranging from $5 \times 10^{-7}$ to $1 \times 10^{-3}$ to identify the best hyperparameters for each pre-trained model[3].

**Evaluation.** We evaluate models at two stages: after pre-training and after SFT. For pre-trained models, we assess zero-shot performance on standard benchmarks, including question answering (ARC-Easy, ARC-Challenge (Clark et al., 2018), OpenBookQA (Mihaylov et al., 2018), BoolQ (Clark et al., 2019)) and commonsense reasoning (HellaSwag (Zellers et al., 2019), PIQA (Bisk et al., 2020), WinoGrande (Sakaguchi et al., 2021)), along with validation loss.

For fine-tuned models, we follow the setup of OLMo (Groeneveld et al., 2024) and evaluate along three key dimensions: instruction-following capability (AlpacaEval (Li et al., 2023)), multi-task language understanding (MMLU (Hendrycks et al., 2021)), and truthfulness (TruthfulQA (Lin et al., 2022)).

To highlight how LR decay affects both pre-training and SFT differently, we present results as relative performance metrics normalized against the best decay-based scheduler for each stage. For pre-training, we report both validation loss and the average accuracy across all zero-shot benchmarks (PT Task Avg). For fine-tuning, we report the average across AlpacaEval, TruthfulQA, and MMLU (SFT Task Avg)[4].

## 3.2 RESULTS

Table 1 shows an inversion in model performance across training stages[5]. For pre-training performance, decay-based schedulers achieve the best performance with $\alpha_{\text{pre}} = 0$. Specifically, Linear and WSD with $\alpha_{\text{pre}} = 0$ achieve the best PT Task Avg scores for the 1B and 8B models, respectively. This result is consistent with existing findings (Bergsma et al., 2025). In contrast, after SFT, WSO achieves the best performance for both model sizes, even though it underperforms decay-based schedulers in pre-training metrics. These results demonstrate that while decay-based schedulers may

---

[2]https://huggingface.co/datasets/allenai/tulu-3-sft-olmo-2-mixture/tree/main

[3]Full details about SFT are provided in Appendix D.

[4]Detailed evaluation settings are provided in Appendix E.

[5]Detailed per-task evaluation results for all models are provided in Appendix F.

Table 2: Relative performance across mid-training (MT) and SFT stages. Values are differences from the best decay-based schedule. WSO throughout both stages yields the best SFT performance.

| Model | (Pre-training) Scheduler | $\alpha_{\text{pre}}$ | $\alpha_{\text{mid}}$ | MT Valid Loss ↓ Δ | MT Task Avg Δ | SFT Task Avg Δ |
|---|---|---|---|---|---|---|
| | Warmup-Stable-Only (WSO) | 1.0 | 1.0 | *+0.062* | *-0.1* | ***+0.8*** |
| | | 1.0 | 0.0 | **+0.000** | **+0.0** | +0.0 |
| | WSD | 0.1 | 1.0 | +0.038 | -1.5 | -0.5 |
| | | 0.1 | 0.0 | +0.047 | -1.7 | -1.3 |
| 1B | Linear | 0.1 | 1.0 | +0.053 | -2.1 | -2.5 |
| | | 0.1 | 0.0 | +0.058 | -3.3 | -3.8 |
| | Cosine | 0.1 | 1.0 | +0.053 | -2.4 | -2.9 |
| | | 0.1 | 0.0 | +0.059 | -3.1 | -3.7 |
| | Warmup-Stable-Only (WSO) | 1.0 | 1.0 | *+0.102* | *-2.1* | ***+1.1*** |
| | | 1.0 | 0.0 | **+0.000** | **+0.0** | -1.4 |
| | WSD | 0.1 | 1.0 | +0.057 | -5.0 | +0.0 |
| | | 0.1 | 0.0 | +0.081 | -5.6 | -1.1 |
| 8B | Linear | 0.1 | 1.0 | +0.067 | -8.3 | -2.2 |
| | | 0.1 | 0.0 | +0.082 | -9.0 | -3.7 |
| | Cosine | 0.1 | 1.0 | +0.068 | -8.0 | -3.5 |
| | | 0.1 | 0.0 | +0.084 | -10.1 | -4.1 |

yield superior performance in terms of pre-training metrics, WSO is more effective in the overall training pipeline, including SFT.

## 4 EXPERIMENT 2: THREE-STAGE (PRE-, MID-, AND POST-TRAINING) SETTING

Recent LLM developments (OLMo et al., 2024; Meta, 2024c) add a mid-training stage between pre-training and post-training, which makes LR scheduling across stages more complex due to the various combinations of pre-training and mid-training LR schedulers. We investigate whether using WSO in both pre-training and mid-training stages yields better performance after SFT than decay-based schedulers.

### 4.1 EXPERIMENTAL SETUP

To investigate the effect of LR scheduling during mid-training, we conduct experiments following a three-stage training pipeline: pre-training, mid-training, and post-training. We systematically vary the LR schedulers in both pre-training and mid-training stages to understand their individual and combined effects on downstream performance. To ensure comparability with recent mid-training work, our setup largely follows OLMo 2 (OLMo et al., 2024), a representative study of mid-training.

**Pre-training Stage.** We pre-train 1B and 8B models using the same architecture and configuration as described in Section 3. We adopt pre-training dataset `olmo-mix-1124` (OLMo et al., 2024) used in OLMo 2. Following standard practice in modern LLM development (Meta, 2024c; OLMo et al., 2024), we employ four LR schedulers with different minimum LR factors, including WSD, Cosine, and Linear schedulers with $\alpha_{\text{pre}} = 0.1$, and additionally WSO.

**Mid-training Stage and Learning Rate Schedules.** Following OLMo 2 (OLMo et al., 2024), we conduct mid-training on the `dolmino-mix-1124` dataset. We investigate the two mid-training strategies shown in Figure 2, with $\alpha_{\text{mid}} = 0.0$ applying further Linear decay following common practice (Meta, 2024c; OLMo et al., 2024), and $\alpha_{\text{mid}} = 1.0$ maintaining a constant LR throughout mid-training[6].

---

[6]Further training configurations of mid-training are provided in Appendix G.

Table 3: Relative performance after over-training (2T tokens). Values are differences ($\Delta$) from the best-performing decay-based scheduler for each metric. Similar to Section 3, WSO achieves the best SFT performance.

| Model | Scheduler | $\alpha_{pre}$ | PT Valid Loss $\downarrow \Delta$ | PT Task Avg $\Delta$ | SFT Task Avg $\Delta$ |
|---|---|---|---|---|---|
|  | Warmup-Stable-Only (WSO) | 1.0 | *+0.048* | *-1.5* | ***+0.7*** |
|  | WSD | 0.1 | +0.004 | **+0.0** | +0.0 |
|  |  | 0.0 | **+0.000** | **+0.0** | -0.3 |
| 1B | Linear | 0.1 | +0.021 | -0.9 | -0.5 |
|  |  | 0.0 | +0.017 | -0.4 | -0.6 |
|  | Cosine | 0.1 | +0.017 | **+0.0** | -0.4 |
|  |  | 0.0 | +0.017 | -1.3 | -0.3 |

Table 4: Relative performance after over-training with mid-training (2T + 500B tokens). Values are differences ($\Delta$) from the best-performing decay-based scheduler for each metric. Similar to Section 4, WSO yields the best SFT performance.

| Model | (Pre-training) Scheduler | $\alpha_{pre}$ | $\alpha_{mid}$ | MT Valid Loss $\downarrow \Delta$ | MT Task Avg $\Delta$ | SFT Task Avg $\Delta$ |
|---|---|---|---|---|---|---|
|  | Warmup-Stable-Only (WSO) | 1.0 | 1.0 | *+0.055* | *-0.3* | ***+1.4*** |
|  | WSD | 1.0 | 0.0 | **+0.000** | -1.6 | -0.5 |
|  |  | 0.1 | 1.0 | +0.033 | **+0.0** | -1.0 |
| 1B |  | 0.1 | 0.0 | +0.038 | -1.7 | -1.2 |
|  | Linear | 0.1 | 1.0 | +0.068 | -2.2 | +0.0 |
|  |  | 0.1 | 0.0 | +0.051 | -2.8 | -0.6 |
|  | Cosine | 0.1 | 1.0 | +0.046 | -1.8 | -0.7 |
|  |  | 0.1 | 0.0 | +0.054 | -2.3 | -1.2 |

**SFT and Evaluation.** For SFT, we follow the configuration described in Section 3. For mid-trained models (before SFT), we evaluate on standard benchmarks to assess the impact of mid-training LR schedulers, following the evaluation suite used in OLMo 2 (OLMo et al., 2024). We select benchmarks that comprehensively assess model capabilities, including reasoning tasks (ARC-Challenge (Clark et al., 2018), HellaSwag (Zellers et al., 2019), WinoGrande (Sakaguchi et al., 2021)), reading comprehension (DROP (Dua et al., 2019)), and mathematical reasoning (GSM8K (Cobbe et al., 2021)). Following SFT, we assess models using an expanded evaluation suite including AlpacaEval (Li et al., 2023) for instruction following, TruthfulQA (Lin et al., 2022) for factual accuracy, GSM8K (Cobbe et al., 2021) for mathematical reasoning, DROP (Dua et al., 2019) for reading comprehension, AGI Eval (Zhong et al., 2024) for general intelligence capabilities, BigBench-Hard (Suzgun et al., 2023) for challenging reasoning tasks, and MMLU for multitask understanding[7]. Similar to Section 3, we present results as relative improvements compared to the best decay-based scheduler.

## 4.2 RESULTS

Table 2 shows an inversion similar to our pre-training findings[8]. For mid-training performance, the decay-based scheduler with $\alpha_{pre} = 1.0$ and $\alpha_{mid} = 0.0$ achieve the best performance. However, SFT performance again shows the opposite trend. WSO achieves the best downstream task performance after SFT, even though it underperforms the best decay-based schedulers in mid-training metrics. Additionally, we find that introducing decay at any stage reduces SFT performance. Notably, for models pre-trained with decay ($\alpha_{pre} = 0.1$), avoiding decay during mid-training ($\alpha_{mid} = 1.0$) improves both mid-training metrics and SFT performance compared to applying decay.

---

[7]The detailed evaluation settings for these benchmarks are described in Appendix E.

[8]Detailed per-task evaluation results for all models are provided in Appendix F.

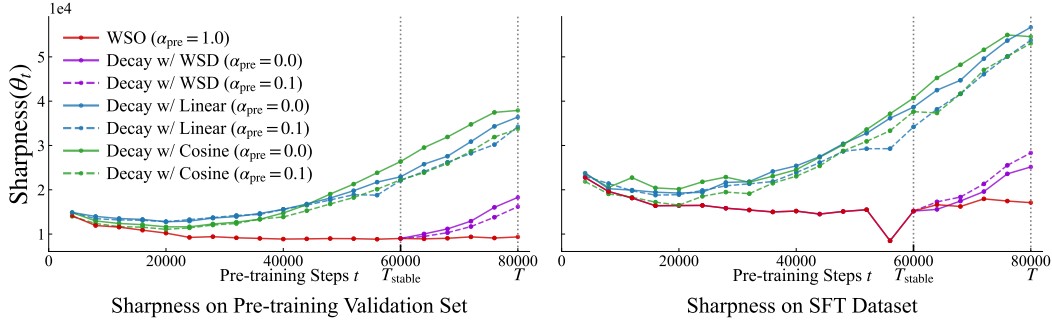

Figure 3: Sharpness($\theta_t$) during pre-training of the 1B model. Vertical line at step $T_{\text{stable}}$ indicating where WSD decays LR. Decay-based schedulers ($\alpha_{\text{pre}} = 0$ or $\alpha_{\text{pre}} = 0.1$) lead to sharper minima, while WSO ($\alpha_{\text{pre}} = 1.0$) maintains flatter landscapes.

These results extend our findings to multi-stage training pipelines, where decay at any stage consistently harms SFT performance. WSO, which maintains constant learning rates throughout both pre-training and mid-training, shows the best performance across the overall training pipeline, including mid-training and SFT.

## 5 EXPERIMENT 3: THREE-STAGE SETTING IN THE OVER-TRAINING

To further probe generality, we evaluate a third regime with a substantially larger training budget. This over-training setting serves as a test of whether the benefits of WSO persist when training on trillions of tokens.

### 5.1 EXPERIMENTAL SETUP

**Pre- and Mid-training.** We pre-train 1B models on 2T tokens, which is approximately $100\times$ the Chinchilla-optimal amount of data for this model size, to evaluate whether WSO maintains its advantages at this data scale. We use the same datasets as in Sections 3 and 4 for pre-training and mid-training, respectively. We investigate the same set of LR schedulers as in Section 3. We additionally conduct mid-training experiments using 500B tokens, following the same experimental setup as in Section 4.

**Evaluation.** We evaluate all LR schedulers using the same methodology as in Sections 3 and 4, measuring performance both after pre-training (or mid-training) and after SFT. Detailed configurations are provided in Appendices C and D.

### 5.2 RESULTS

Tables 3 and 4 confirm that the inversion observed in Sections 3 and 4 persists even in the over-training regime using 2T tokens. Across all investigated schedulers, WSO ($\alpha_{\text{pre}} = 1.0$) consistently yields worse intermediate metrics but superior SFT performance compared to decay-based schedulers. Similar to our earlier findings, decay-based schedulers achieve better pre-training and mid-training metrics, yet WSO outperforms them after SFT. This pattern holds both for single-stage over-training (Table 3) and when combined with mid-training (Table 4), demonstrating that the benefits of WSO are robust across different data scales and training configurations.

## 6 UNDERSTANDING ADAPTABILITY THROUGH LOSS LANDSCAPE GEOMETRY

To understand why models trained with WSO achieve superior SFT performance, we analyze the loss landscape geometry throughout pre-training phase. As suggested in the transfer learning lit-

erature (Ju et al., 2022; Liu et al., 2023), we focus on sharpness as a key geometric property that characterizes the curvature of the loss landscape around converged parameters.

The relation between lower sharpness and better SFT performance stems from how models respond to parameter updates during fine-tuning. When the parameters of the model lie in a flatter region of the loss landscape, which corresponds to lower sharpness, the model demonstrates superior adaptability to downstream tasks (Foret et al., 2021; Li et al., 2025). The intuition is that the performance of the model remains stable during the parameter updates of SFT. A model in a flat landscape experiences less fluctuation in its loss value when its parameters are updated, which translates to more stable performance. This characteristic is believed to confer higher adaptability, as the model can incorporate new data without compromising its pre-trained capabilities (Andriushchenko et al., 2023).

There are several ways to quantify sharpness, such as the largest eigenvalue of the Hessian (capturing the most curved direction) or the trace of the Hessian (capturing the average curvature) (Dinh et al., 2017; Kaur et al., 2023). Following established practice in optimization and generalization studies (Ju et al., 2022; Liu et al., 2023), we adopt the trace as our sharpness measure, since it provides a scalar summary of curvature across all parameter dimensions.

**Definition 6.1** (Sharpness). Let $\mathcal{L}(\theta_t; \mathcal{D})$ denote the loss function evaluated on dataset $\mathcal{D}$ with model parameters $\theta_t \in \mathbb{R}^d$. At training step $t$, the sharpness of the loss landscape at parameters $\theta_t$ is defined as the trace of the Hessian matrix:

$$\text{Sharpness}(\theta_t) = \text{Tr}(\mathbf{H}_{\mathcal{L}}(\theta_t)) = \sum_{i=1}^{d} \frac{\partial^2 \mathcal{L}(\theta_t; \mathcal{D})}{\partial \theta_i^2} \tag{6}$$

where $\mathbf{H}_{\mathcal{L}}(\theta_t) \in \mathbb{R}^{d \times d}$ is the Hessian matrix of the loss with respect to the parameters at $\theta_t$.

Since computing the full Hessian trace is computationally prohibitive for billion-parameter models, we employ Hutchinson's unbiased estimator (Hutchinson, 1989; Liu et al., 2024b). This method requires only Hessian-vector products, which can be efficiently computed through automatic differentiation. Details of our sampling procedure and computational details are provided in Appendix H.

We measure sharpness throughout pre-training on validation sets from both the pre-training dataset and the SFT dataset. Figure 3 shows the sharpness for the 1B model from Section 3. We illustrate a vertical line at step $T_{\text{stable}}$ to indicate the point at which WSD decays LR. The figure reveals distinct patterns across schedulers. Specifically, Cosine and Linear schedulers exhibit steadily increasing sharpness as the LR decays, while WSD shows a rise during its decay phase. In contrast, WSO maintains lower sharpness. Across both datasets, models with decaying LRs converge to regions with about 2–3× higher sharpness compared to WSO models. Flatter regions obtained by WSO allow more flexible parameter adaptation during SFT, enabling better downstream performance.

**Correlation between sharpness and downstream adaptability.** To provide empirical evidence linking the loss landscape to downstream adaptability, we analyze the correlation between the sharpness of pre-trained models and their subsequent SFT performance. Figure 4 presents the SFT performance plotted against the sharpness measured on the pretraining validation set for the 1B model across all investigated learning rate schedulers. The analysis reveals a negative correlation (Pearson $r = -0.709$) between the sharpness of the minima and the model's performance after SFT. The WSO scheduler ($\alpha_{\text{pre}} = 1.0$) resides in the low-sharpness, high-performance region, while

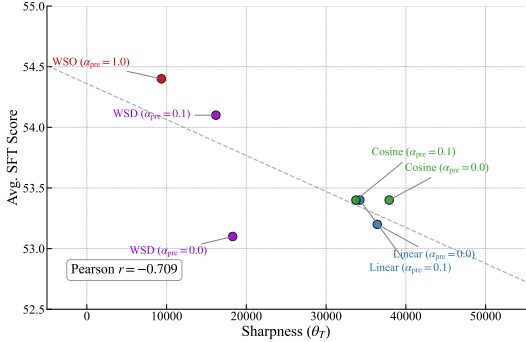

Figure 4: Pre-training sharpness negatively correlates with downstream SFT performance.

decay-based schedulers converge to sharper minima with lower SFT scores. While the sample size is limited, this pattern is consistent with our hypothesis that preserving flatter minima during pretraining enhances the model's adaptability.

## 7    RELATED WORK

**Learning Rate Scheduling in LLM Training.**    LR decay has been considered effective for LLM pre-training, with Cosine decay remaining the de facto standard (Kaplan et al., 2020; Hoffmann et al., 2022; Touvron et al., 2023b). Recent large-scale studies advocate for even more aggressive decay, showing that Linear decay to zero achieves lower pre-training loss in compute-optimal settings (Bergsma et al., 2025). Warmup-Stable-Decay (WSD) delays decay until the final phase of training (Hu et al., 2024), while theoretical analysis suggests that decay may confine models to narrow loss valleys (Wen et al., 2025). Some methods attempt to avoid the decay phase through checkpoint averaging (Sanyal et al., 2024) or model merging (Tian et al., 2025). Jin et al. (2023) investigated learning rate tuning strategies within individual training phases, but did not examine how pre-training LR choices propagate through to post-training performance. While these studies have advanced our understanding of LR scheduling within a single training phase, they share a common limitation: evaluation is restricted to the phase in which the schedule is applied, without considering the downstream consequences for subsequent training stages such as SFT.

**Relationship to Continual Pre-training and Fine-tuning Studies.**    Recent work on continual pre-training (CPT) has explored LR scheduling for domain adaptation. Wang et al. (2025) showed that models with higher loss potential achieve lower CPT validation loss and advocated for releasing high loss potential versions to facilitate downstream tasks. Wang et al. (2024) proposed a path-switching paradigm for LR scheduling in model version updates, though their experimental setup still applies LR decay before performing SFT. Tissue et al. (2024) introduced a scaling law describing loss dynamics in relation to learning rate annealing; however, they explicitly note that post-training scenarios involving distribution shift are out of scope. These CPT studies focus on settings where the objective function remains language modeling, leaving the impact on SFT unexplored. Meanwhile, a growing body of work has examined the gap between pre-training quality and downstream performance more broadly. Sun & Dredze (2025) showed that stronger pre-training performance does not necessarily translate to superior fine-tuning outcomes, and Springer et al. (2025) demonstrated that over-trained models become harder to fine-tune. These findings collectively motivate our investigation, as LR schedulers chosen to optimize potentially unreliable pre-training metrics may not be optimal for the overall training pipeline. Our work identifies LR decay as a specific factor that improves pre-training metrics at the cost of downstream adaptability.

**Adaptability and Loss Landscape Geometry.**    Early work showed that parameters in flatter loss regions generalize better than those in sharp minima (Keskar et al., 2017), motivating sharpness-aware minimization (Foret et al., 2021) and stochastic weight averaging (Izmailov et al., 2018). Recent theoretical advances explain WSD through a river valley loss landscape perspective (Wen et al., 2025), where the stable phase explores along the valley floor while the decay phase converges toward the center. Concurrent work confirmed that sharpness increase during decay is universal across architectures (Belloni et al., 2025). Flat-minima optimizers work well under distribution shift (Kaddour et al., 2022), a property that extends to the pre-training/fine-tuning paradigm, where the fine-tuning data distribution differs substantially from pre-training (Ju et al., 2022; Liu et al., 2023). While prior work focused on understanding sharpness dynamics during pre-training (Belloni et al., 2025; Wen et al., 2025), we demonstrate how these dynamics concretely impact SFT performance, showing that WSO preserves flatness and enhances downstream adaptability.

## 8    CONCLUSION

In this study, we investigated the effectiveness of LR schedulers, which have been widely reported as effective for pre-training, in practical scenarios with a focus on post-training performance. In particular, we examine Warmup-Stable-Only (WSO), which removes the decay phase from WSD. Experimental results show that WSO consistently outperforms decay-based schedulers in downstream tasks after SFT across standard pre-training, mid-training, and over-training regimes. Loss landscape analysis further reveals that WSO preserves flatter minima, explaining its superior adaptability. WSO is simple to apply and yields improved post-training performance, making it a promising alternative for constructing more portable models. We also recommend releasing LLMs trained with WSO so that practitioners can benefit from their adaptability.

## ETHICS STATEMENT

This work investigates learning rate scheduling for LLM training to improve downstream adaptability. While our methods may provide new findings on LR scheduling on pre-training, we acknowledge the broader implications of advancing LLM capabilities. We encourage responsible deployment with appropriate safety measures during post-training. We exclusively used publicly available datasets for pre-training, supervised fine-tuning, and evaluation. Moreover, we developed the language models entirely from scratch, avoiding the use of any publicly available models to ensure reproducibility.

## REPRODUCIBILITY STATEMENT

To ensure reproducibility of our results, we provide comprehensive experimental details throughout the paper and appendices. Model architectures for both 1B and 8B parameter models are specified in Appendix A, including all layer configurations and attention mechanisms. All pre-training hyperparameters, including optimizer settings, batch sizes, and training steps, are detailed in Appendix C. The supervised fine-tuning configuration, including the learning rate sweep range and evaluation protocols, is described in Appendix D. Our sharpness computation methodology using Hutchinson's estimator is fully specified in Appendix H. We use publicly available datasets (FineWeb-Edu, olmo-mix-1124, dolmino-mix-1124, and Tulu-3 SFT mixture) and standard evaluation benchmarks, with detailed evaluation settings provided in Appendix E. Full numerical results for all experiments are reported in Appendix F to facilitate comparison and validation.

### ACKNOWLEDGMENTS

This work was partly supported by JST Moonshot R&D Grant Number JPMJMS2011-35 (fundamental research).

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

Table 5: Model configurations for the 1B and 8B models.

| Configuration | 1B | 8B |
|---|---|---|
| Hidden dimension | 2048 | 4096 |
| FFN dimension | 8192 | 14336 |
| Number of layers | 16 | 32 |
| Number of heads | 32 | 32 |
| Number of KV heads | 8 | 8 |
| Head dimension | 64 | 128 |
| Vocabulary size | 128256 | 128256 |
| RoPE $\theta$ | 10000 | 10000 |
| RMS norm $\epsilon$ | $10^{-5}$ | $10^{-5}$ |
| Activation function | SwiGLU | SwiGLU |

## A  MODEL ARCHITECTURE

We provide detailed specifications for the models used in our experiments. Both the 1B and 8B models follow the Llama 3 architecture (Meta, 2024c), employing RMSNorm, SwiGLU activation, and Rotary Position Embeddings. We use the Llama 3 tokenizer with a vocabulary size of 128,256 tokens for all models.

## B  LEARNING RATE SCHEDULER FORMULATIONS

We provide the complete formulations for the WSD, Cosine, and Linear LR schedulers used in our experiments.

**WSD Schedule:** After warmup, the LR remains constant until $T_{\text{stable}}$, then decays linearly to $\alpha_{\text{pre}} \cdot \eta_{\max}$ at step $T$:

$$
\eta^{\text{WSD}}(t, \alpha_{\text{pre}}) = \begin{cases} \eta_{\max} \cdot \frac{t}{T_{\text{warmup}}} & t \leq T_{\text{warmup}} \\ \eta_{\max} & T_{\text{warmup}} < t \leq T_{\text{stable}} \\ \eta_{\max} \cdot \left( (1 - \alpha_{\text{pre}}) \cdot \frac{T_{\text{pre}} - t}{T_{\text{pre}} - T_{\text{stable}}} + \alpha_{\text{pre}} \right) & T_{\text{stable}} < t \leq T_{\text{pre}} \end{cases}
\tag{7}
$$

**WSO Schedule:** Obtained by setting $\alpha_{\text{pre}} = 1$ in WSD. After warmup, the LR stays constant:

$$
\eta^{\text{WSO}}(t, \alpha_{\text{pre}}) = \begin{cases} \eta_{\max} \cdot \frac{t}{T_{\text{warmup}}} & t \leq T_{\text{warmup}} \\ \eta_{\max} & T_{\text{warmup}} < t \leq T_{\text{pre}} \end{cases}
\tag{8}
$$

**Cosine Schedule:** After warmup, the LR follows a Cosine decay to $\alpha_{\text{pre}} \cdot \eta_{\max}$:

$$
\eta^{\text{Cosine}}(t, \alpha_{\text{pre}}) = \begin{cases} \eta_{\max} \cdot \frac{t}{T_{\text{warmup}}} & t \leq T_{\text{warmup}} \\ \eta_{\max} \cdot \left( \alpha_{\text{pre}} + \frac{1 - \alpha_{\text{pre}}}{2} \left( 1 + \cos \left( \frac{t - T_{\text{warmup}}}{T_{\text{pre}} - T_{\text{warmup}}} \cdot \pi \right) \right) \right) & t > T_{\text{warmup}} \end{cases}
\tag{9}
$$

**Linear Schedule:** After warmup, the LR decays linearly to $\alpha_{\text{pre}} \cdot \eta_{\max}$:

$$
\eta^{\text{Linear}}(t, \alpha_{\text{pre}}) = \begin{cases} \eta_{\max} \cdot \frac{t}{T_{\text{warmup}}} & t \leq T_{\text{warmup}} \\ \eta_{\max} \cdot \left( (1 - \alpha_{\text{pre}}) \cdot \frac{T_{\text{pre}} - t}{T_{\text{pre}} - T_{\text{warmup}}} + \alpha_{\text{pre}} \right) & t > T_{\text{warmup}} \end{cases}
\tag{10}
$$

All the schedulers use the same warmup phase as described in Section 2.4, and their decay is controlled by the minimum LR factor $\alpha_{\text{pre}} \in [0.0, 1.0]$.

**Mid-training LR Scheduling.**   In the mid-training stage, we extend the pre-training learning rate schedulers. The mid-training learning rate at time step $t$ is defined as:

Table 6: Pre-training hyperparameters for 1B and 8B models. The WSD stable ratio $\rho = 0.75$ means the LR remains stable for 75% of training after warmup, with decay occurring in the final 25% when $\alpha_{\text{pre}} < 1$.

| Hyperparameter | 1B | 8B |
|---|---|---|
| *Training Configuration* | | |
| Total training steps | 80,000 | 80,000 |
| Total tokens | 350B | 500B |
| Batch size (tokens) | 4,194,304 | 12,582,912 |
| Sequence length | 2,048 | 2,048 |
| *Optimizer (AdamW)* | | |
| Max LR ($\eta_{\text{max}}$) | $3 \times 10^{-4}$ | $3 \times 10^{-4}$ |
| Weight decay | 0.1 | 0.1 |
| Adam $\beta_1$ | 0.9 | 0.9 |
| Adam $\beta_2$ | 0.95 | 0.95 |
| Adam $\epsilon$ | $1 \times 10^{-8}$ | $1 \times 10^{-8}$ |
| Gradient clipping | 1.0 | 1.0 |
| *LR Schedule* | | |
| Warmup steps ($T_{\text{warmup}}$) | 1,000 | 1,000 |
| WSD stable ratio ($\rho$) | 0.75 | 0.75 |
| Min LR factor ($\alpha_{\text{pre}}$) | {0.0, 0.1, 1.0} | {0.0, 0.1, 1.0} |
| *Other* | | |
| Precision | bfloat16 | bfloat16 |

Table 7: Over-training configuration for the 1B model trained on 2T tokens. All other hyperparameters are identical to those in Table 6.

| Hyperparameter | Value |
|---|---|
| *Training Configuration* | |
| Total training steps | 120,000 |
| Total tokens | 2T |
| Batch size (tokens) | 16,777,216 |

$$\eta^{\text{Scheduler}}(t, \alpha_{\text{pre}}, \alpha_{\text{mid}}) = \eta^{\text{Scheduler}}(T_{\text{pre}}, \alpha_{\text{pre}}) \cdot \left( (1 - \alpha_{\text{mid}}) \cdot \frac{T_{\text{pre}} + T_{\text{mid}} - t}{T_{\text{mid}}} + \alpha_{\text{mid}} \right) \quad (11)$$

for $t \in [T_{\text{pre}} + 1, T_{\text{pre}} + T_{\text{mid}}]$, where $T_{\text{pre}}$ is the total number of pre-training steps and $T_{\text{mid}}$ is the total number of mid-training steps.

## C  PRE-TRAINING HYPERPARAMETERS

We provide detailed hyperparameters used for pre-training our models in Table 6. All experiments use the AdamW optimizer (Loshchilov & Hutter, 2019) with mixed precision. For over-training experiments, we modify the training duration as shown in Table 7, where the 1B model is trained for 120,000 steps to process 2T tokens and set different batch sizes while maintaining the other hyperparameters in Table 6.

## D  SFT CONFIGURATION

We performed supervised fine-tuning for all models using the Tulu-3 SFT mixture dataset. Since the official dataset does not provide a predefined train-validation split, we create our own using a 9:1 ratio for training and validation, respectively. We perform full parameter training for all models. Table 8 presents the hyperparameters used in our experiments.

Table 8: SFT hyperparameters used in our experiments. We perform a sweep over the specified LRs and select the best value based on AlpacaEval performance.

| Hyperparameter | Value |
|---|---|
| LR | $5.0 \times 10^{-7}, 1.0 \times 10^{-6}, 5.0 \times 10^{-6}, 1.0 \times 10^{-5}, 5.0 \times 10^{-5}, 1.0 \times 10^{-4}, 5.0 \times 10^{-4}, 1.0 \times 10^{-3}$ |
| Global Batch size | 128 |
| LR scheduler | Cosine with warmup |
| Minimum LR | 0 |
| Optimizer | AdamW |
| Weight decay | 0.0 |
| Gradient clipping | 1.0 |
| Warmup steps | 100 |
| Epochs | 1 |
| Training precision | bfloat16 |

## E  EVALUATION DETAILS

For pre-trained models, all benchmarks are evaluated in a zero-shot setting.

For mid-trained models (before SFT), we evaluate on standard benchmarks following the evaluation suite used in OLMo 2 (OLMo et al., 2024). We assess reasoning capabilities using **ARC-Challenge** (Clark et al., 2018), **HellaSwag** (Zellers et al., 2019), and **WinoGrande** (Sakaguchi et al., 2021). Reading comprehension is evaluated with **DROP** (Dua et al., 2019) using 5-shot prompting, while mathematical reasoning is assessed using **GSM8K** (Cobbe et al., 2021) with 8-shot chain-of-thought (CoT) prompting.

For SFT models, we use the following evaluation configurations. For **AlpacaEval**, following Springer et al. (2025), rather than comparing against GPT-4o, where the win rates would be uniformly low, we use a reference model of the same architecture to better distinguish performance differences between LR schedules. Specifically, we use the WSO model with $\alpha_{\text{pre}} = 1.0$, fine-tuned with the lowest LR from our sweep ($5 \times 10^{-7}$) as our reference, ensuring stable and meaningful comparisons within each model scale. Evaluations are performed by Llama-3-70B-Instruct. For **MMLU** (5-shot), evaluation covers 57 subjects spanning STEM, humanities, social sciences, and other domains. For **TruthfulQA**, we use the standard evaluation protocol. After mid-training and SFT, we additionally evaluate on **GSM8K** (1-shot), **DROP** (5-shot), **AGI Eval** (Zhong et al., 2024) (3-shot), and **BigBench-Hard** (Suzgun et al., 2023) (3-shot with CoT).

## F  FULL EVALUATION RESULTS

This section provides complete per-task evaluation results for all pre-trained and fine-tuned models across different LR schedules. While the main text presents aggregated metrics and relative performance comparisons, here we report the absolute performance values for each individual benchmark.

### F.1  PRE-TRAINING EVALUATION RESULTS

Table 9 presents comprehensive zero-shot evaluation results for all pre-trained models across different LR schedules.

#### F.1.1  PRE-TRAINING EVALUATION RESULTS IN OVER-TRAINING

Table 10 shows that, also in the over-training regime with 2T tokens, the Cosine scheduler with decay achieves slightly better zero-shot task performance and lower validation loss compared to WSO.

### F.2  SFT EVALUATION RESULTS

We select the best learning rate for each pre-trained model based on AlpacaEval performance, as the primary objective of SFT is to enhance instruction-following capabilities. Selecting hyperparameters based on such as validation loss does not necessarily yield better downstream task performance,

Table 9: Pre-training evaluation results. Models with more decay ($\alpha_{\text{pre}} = 0$) generally achieve lower validation loss, but not always better zero-shot task performance.

| Model | Scheduler | $\alpha_{\text{pre}}$ | Valid Loss ↓ | ARC-e | ARC-c | BoolQ | Hella | OBQA | PIQA | Wino | Avg. |
|---|---|---|---|---|---|---|---|---|---|---|---|
| 1B | Warmup-Stable-Only (WSO) | 1.0 | 2.431 | 70.8 | 42.2 | 62.0 | 56.3 | 45.4 | 70.8 | 58.5 | 58.0 |
| | WSD | 0.1 | 2.364 | 72.0 | 40.0 | 62.1 | 57.4 | 46.4 | **72.5** | 57.1 | 58.2 |
| | | 0.0 | **2.360** | 72.2 | 39.7 | 63.7 | 57.6 | 45.6 | 72.2 | **58.6** | 58.5 |
| | Linear | 0.1 | 2.380 | 70.3 | 42.6 | 63.2 | 55.6 | 45.2 | 71.6 | 55.7 | 57.7 |
| | | 0.0 | 2.376 | 74.4 | 43.4 | 65.7 | 58.4 | 47.4 | 70.9 | 57.5 | **59.7** |
| | Cosine | 0.1 | 2.379 | 71.1 | **43.6** | **66.5** | **59.9** | 47.8 | 71.7 | 56.3 | 59.6 |
| | | 0.0 | 2.376 | **74.6** | 41.9 | 50.7 | 58.5 | **48.4** | 71.0 | 55.4 | 57.2 |
| 8B | Warmup-Stable-Only (WSO) | 1.0 | 2.119 | 79.4 | 52.6 | 69.1 | 69.1 | 52.8 | 76.3 | 64.5 | 66.3 |
| | WSD | 0.1 | 2.011 | 80.4 | 52.8 | 69.1 | 72.6 | 53.2 | 75.9 | 64.0 | 66.9 |
| | | 0.0 | 2.005 | **81.0** | **53.0** | 67.2 | **72.9** | **54.2** | 76.3 | **65.0** | **67.1** |
| | Linear | 0.1 | 2.004 | 79.4 | 53.7 | 64.1 | 71.2 | 50.4 | 75.0 | 62.4 | 65.2 |
| | | 0.0 | **1.992** | 76.6 | 48.2 | 71.1 | 71.5 | 53.6 | 74.9 | 61.3 | 65.3 |
| | Cosine | 0.1 | 2.001 | 76.3 | 47.6 | **71.3** | 71.5 | 52.4 | 74.3 | 60.9 | 64.9 |
| | | 0.0 | 2.000 | 74.2 | 46.8 | 71.7 | 71.4 | 52.6 | **76.3** | 60.8 | 64.8 |

Table 10: Pre-training evaluation results for over-trained 1B models (2T tokens).

| Model | Scheduler | $\alpha_{\text{pre}}$ | Valid Loss ↓ | ARC-e | ARC-c | BoolQ | Hella | OBQA | PIQA | Wino | Avg. |
|---|---|---|---|---|---|---|---|---|---|---|---|
| 1B | Warmup-Stable-Only (WSO) | 1.0 | 2.625 | 74.4 | 43.3 | 59.7 | 63.5 | 48.6 | 73.2 | 62.0 | 60.7 |
| | WSD | 0.1 | 2.582 | **75.4** | 45.8 | 60.0 | **66.6** | 50.0 | **74.7** | 62.7 | **62.2** |
| | | 0.0 | **2.578** | 75.3 | **46.8** | 59.2 | 66.2 | 50.4 | 74.4 | **63.0** | **62.2** |
| | Linear | 0.1 | 2.599 | 73.4 | 45.6 | 64.7 | 65.4 | 48.0 | 73.2 | 58.8 | 61.3 |
| | | 0.0 | 2.595 | 73.9 | 44.4 | **66.6** | 65.2 | 49.2 | 73.6 | 59.8 | 61.8 |
| | Cosine | 0.1 | 2.595 | 72.9 | 44.1 | 65.7 | 64.9 | **52.0** | 74.0 | 61.5 | **62.2** |
| | | 0.0 | 2.595 | 73.7 | 44.4 | 64.4 | 64.0 | 45.8 | 72.7 | 61.1 | 60.9 |

which is consistent with our main finding that lower pre-training loss does not guarantee better post-SFT performance. We apply an identical learning rate sweep to all pre-trained models, ensuring that no scheduler receives a selective advantage. Table 11 shows the selected learning rates for each model.

Table 12 shows performance after SFT across different pre-training schedules. Models pre-trained with WSO or moderate decay ($\alpha_{\text{pre}} = 0.1$) often achieve comparable or better downstream performance than those with aggressive decay ($\alpha_{\text{pre}} = 0.0$), despite having worse pre-training metrics.

### F.2.1 SFT Evaluation Results in Over-training

Table 13 shows the learning rates selected for each over-trained model. Table 14 demonstrates that even after over-training with 2T tokens, WSO achieves superior SFT performance compared to decay-based schedulers.

### F.3 Mid-training Evaluation Results

Table 15 presents evaluation results after the mid-training stage.

### F.3.1 Mid-training Evaluation Results in Over-training

Table 16 shows that after over-training and mid-training, WSO achieves superior overall performance despite having nearly identical validation loss.

### F.4 SFT Evaluation Results After Mid-training

Table 17 shows the optimal learning rates selected for each pre-trained model based on AlpacaEval performance.

Table 11: SFT learning rates selected for each pre-trained model based on AlpacaEval performance.

| Model | Scheduler | $\alpha_{\text{pre}}$ | Selected SFT LR |
|---|---|---|---|
| 1B | Warmup-Stable-Only (WSO) | 1.0 | $3 \times 10^{-4}$ |
| | WSD | 0.1 | $1 \times 10^{-4}$ |
| | | 0.0 | $1 \times 10^{-4}$ |
| | Linear | 0.1 | $1 \times 10^{-4}$ |
| | | 0.0 | $1 \times 10^{-4}$ |
| | Cosine | 0.1 | $1 \times 10^{-4}$ |
| | | 0.0 | $1 \times 10^{-4}$ |
| 8B | Warmup-Stable-Only (WSO) | 1.0 | $3 \times 10^{-4}$ |
| | WSD | 0.1 | $3 \times 10^{-4}$ |
| | | 0.0 | $1 \times 10^{-4}$ |
| | Linear | 0.1 | $1 \times 10^{-4}$ |
| | | 0.0 | $1 \times 10^{-4}$ |
| | Cosine | 0.1 | $1 \times 10^{-4}$ |
| | | 0.0 | $3 \times 10^{-5}$ |

Table 12: SFT evaluation results. Models pre-trained with WSO achieve the best downstream performance.

| Model | Scheduler | $\alpha_{\text{pre}}$ | AlpacaEval | TruthfulQA | MMLU | Avg. |
|---|---|---|---|---|---|---|
| 1B | Warmup-Stable-Only (WSO) | 1.0 | **84.0** | **43.4** | 35.9 | **54.4** |
| | WSD | 0.1 | 83.9 | 41.9 | 36.6 | 54.1 |
| | | 0.0 | 82.3 | 40.2 | **36.7** | 53.1 |
| | Linear | 0.1 | 82.0 | 42.0 | 36.3 | 53.4 |
| | | 0.0 | 82.4 | 41.7 | 35.6 | 53.2 |
| | Cosine | 0.1 | 83.6 | 41.0 | 35.5 | 53.4 |
| | | 0.0 | 83.6 | 41.0 | 35.6 | 53.4 |
| 8B | Warmup-Stable-Only (WSO) | 1.0 | **79.7** | **42.5** | 42.7 | **55.0** |
| | WSD | 0.1 | 77.1 | 40.8 | 41.4 | 53.1 |
| | | 0.0 | 77.3 | 39.9 | **43.7** | 53.6 |
| | Linear | 0.1 | 76.4 | 41.4 | 42.1 | 53.3 |
| | | 0.0 | 78.4 | 40.6 | 42.8 | 53.9 |
| | Cosine | 0.1 | 78.6 | 39.9 | 42.3 | 53.6 |
| | | 0.0 | 77.8 | 40.3 | 43.3 | 53.8 |

Table 18 shows SFT performance after mid-training. WSO during mid-training ($\alpha_{\text{mid}} = 1.0$) generally achieves better SFT performance compared to those with decay ($\alpha_{\text{mid}} = 0.0$).

### F.5 SFT Evaluation Results After Over-training with Mid-training

Table 19 shows the selected learning rates for each over-trained model, and Table 20 shows that WSO achieves superior SFT performance compared to decay-based schedulers.

## G Mid-training Configuration Details

We provide the detailed configuration used for mid-training experiments in Table 21. Other hyperparameters are the same as the configurations of pre-training in Table 6 Mid-training is conducted on the `dolmino-mix-1124` dataset, which consists of diverse high-quality data sources.

Table 13: SFT learning rates selected for each over-trained 1B model based on AlpacaEval performance.

| Model | Scheduler | $\alpha_{\text{pre}}$ | Selected SFT LR |
|---|---|---|---|
| 1B | Warmup-Stable-Only (WSO) | 1.0 | $1 \times 10^{-4}$ |
| | WSD | 0.1 | $3 \times 10^{-5}$ |
| | | 0.0 | $3 \times 10^{-5}$ |
| | Linear | 0.1 | $3 \times 10^{-5}$ |
| | | 0.0 | $3 \times 10^{-5}$ |
| | Cosine | 0.1 | $1 \times 10^{-5}$ |
| | | 0.0 | $1 \times 10^{-4}$ |

Table 14: SFT evaluation results for over-trained 1B models (pre-trained on 2T tokens).

| Model | Scheduler | $\alpha_{\text{pre}}$ | AlpacaEval | TruthfulQA | MMLU | Avg. |
|---|---|---|---|---|---|---|
| 1B | Warmup-Stable-Only (WSO) | 1.0 | **78.1** | **38.7** | **34.5** | **50.4** |
| | WSD | 0.1 | 77.2 | 38.3 | 33.6 | 49.7 |
| | | 0.0 | 76.0 | 38.4 | 33.7 | 49.4 |
| | Linear | 0.1 | 75.6 | 37.8 | 34.2 | 49.2 |
| | | 0.0 | 75.5 | 37.9 | 33.9 | 49.1 |
| | Cosine | 0.1 | 76.0 | 37.9 | 33.9 | 49.3 |
| | | 0.0 | 76.4 | 37.9 | 33.9 | 49.4 |

Additionally, we provide the detailed hyperparameters used for mid-training in over-training settings in Section 5 in Table 22

## H SHARPNESS COMPUTATION DETAILS

We compute the sharpness (Hessian trace) using Hutchinson's stochastic trace estimator (Hutchinson, 1989), which provides an unbiased estimate through random vector sampling. For a Hessian matrix $\mathbf{H}$, the trace is estimated as:

$$\text{Tr}(\mathbf{H}) \approx \frac{1}{N} \sum_{i=1}^{N} \mathbf{z}_i^T \mathbf{H} \mathbf{z}_i \tag{12}$$

where $\mathbf{z}_i$ are random vectors sampled from a Rademacher distribution (i.e., each element is $\pm 1$ with equal probability).

**Implementation Details.** We compute Hessian-vector products using automatic differentiation, which allows efficient computation without explicitly constructing the full Hessian matrix.

Table 23 shows computation configurations for Hutchinson's estimator. We measure sharpness at regular intervals throughout pre-training (every 4,000 steps) on held-out validation sets from both the pre-training dataset and the SFT dataset to understand how the loss landscape geometry evolves during training.

Table 15: Mid-training evaluation results in Section 4

| Model | Pre-training Scheduler | $\alpha_{pre}$ | $\alpha_{mid}$ | Valid Loss ↓ | ARC-C | HellaSwag | WinoGrande | DROP | GSM8K | Avg. |
|---|---|---|---|---|---|---|---|---|---|---|
| | Warmup-Stable-Only (WSO) | 1.0 | 1.0 | 2.335 | **47.0** | 60.5 | 58.6 | 23.9 | 20.4 | 42.1 |
| | WSD | 1.0 | 0.0 | 2.273 | 45.0 | 61.1 | 60.4 | 23.3 | **21.1** | **42.2** |
| | | 0.1 | 0.0 | 2.320 | 45.0 | **62.0** | **60.7** | 23.8 | 11.0 | 40.5 |
| 1B | | 0.1 | 1.0 | **2.310** | 45.1 | 60.7 | 59.8 | **24.5** | 13.0 | 40.6 |
| | Cosine | 0.1 | 0.0 | 2.332 | 43.8 | 61.4 | 59.5 | 20.2 | 10.7 | 39.1 |
| | | 0.1 | 1.0 | 2.326 | 44.3 | 60.7 | 59.7 | 21.4 | 12.8 | 39.8 |
| | Linear | 0.1 | 0.0 | 2.330 | 43.0 | 60.3 | 60.5 | 19.6 | 11.0 | 38.9 |
| | | 0.1 | 1.0 | 2.325 | 43.2 | 60.3 | 60.1 | 23.6 | 13.3 | 40.1 |
| | Warmup-Stable-Only (WSO) | 1.0 | 1.0 | 2.009 | 64.9 | 75.4 | 69.4 | 49.7 | 52.8 | 62.4 |
| | WSD | 1.0 | 0.0 | **1.907** | **69.7** | 77.9 | 70.6 | **50.6** | **53.9** | **64.5** |
| | | 0.1 | 0.0 | 1.988 | 61.4 | **80.0** | **71.1** | 42.6 | 39.7 | 59.0 |
| 8B | | 0.1 | 1.0 | 1.964 | 62.4 | 79.4 | 71.0 | 42.4 | 42.4 | 59.5 |
| | Cosine | 0.1 | 0.0 | 1.991 | 54.3 | 77.0 | 69.7 | 35.4 | 36.0 | 54.5 |
| | | 0.1 | 1.0 | 1.975 | 57.1 | 77.5 | 69.1 | 38.6 | 40.3 | 56.5 |
| | Linear | 0.1 | 0.0 | 1.989 | 55.5 | 77.3 | 71.0 | 36.2 | 37.7 | 55.5 |
| | | 0.1 | 1.0 | 1.974 | 56.7 | 77.5 | 69.9 | 36.6 | 40.3 | 56.2 |

Table 16: Mid-training evaluation results for over-trained 1B models (pre-trained on 2T tokens, mid-trained on 500B tokens).

| Model | Pre-training Scheduler | $\alpha_{pre}$ | $\alpha_{mid}$ | Valid Loss ↓ | ARC-C | HellaSwag | WinoGrande | DROP | GSM8K | Avg. |
|---|---|---|---|---|---|---|---|---|---|---|
| | Warmup-Stable-Only (WSO) | 1.0 | 1.0 | 2.254 | 46.7 | 61.3 | 60.4 | **27.0** | **23.1** | 43.7 |
| | WSD | 1.0 | 0.0 | **2.199** | 47.1 | **65.2** | 62.2 | 23.7 | 13.7 | 42.4 |
| | | 0.1 | 0.0 | 2.237 | 47.1 | **65.2** | 62.2 | 23.4 | 13.7 | 42.3 |
| 1B | | 0.1 | 1.0 | 2.231 | 47.4 | 65.7 | **62.6** | 25.3 | 19.1 | **44.0** |
| | Cosine | 0.1 | 0.0 | 2.253 | 46.0 | 65.1 | 62.3 | 23.8 | 11.4 | 41.7 |
| | | 0.1 | 1.0 | 2.245 | 43.5 | 64.7 | 62.1 | 25.9 | 14.8 | 42.2 |
| | Linear | 0.1 | 0.0 | 2.250 | **47.2** | 63.4 | 59.4 | 20.9 | 15.4 | 41.3 |
| | | 0.1 | 1.0 | 2.267 | 45.6 | 63.3 | 60.1 | 21.4 | 18.7 | 41.8 |

Table 17: SFT learning rates selected for each model configuration based on AlpacaEval performance.

| Model | Scheduler | $\alpha_{pre}$ | $\alpha_{mid}$ | Selected SFT LR |
|---|---|---|---|---|
| | Warmup-Stable-Only (WSO) | 1.0 | 1.0 | $3 \times 10^{-4}$ |
| | WSD | 1.0 | 0.0 | $3 \times 10^{-5}$ |
| | | 0.1 | 1.0 | $1 \times 10^{-4}$ |
| 1B | | 0.1 | 0.0 | $3 \times 10^{-5}$ |
| | Linear | 0.1 | 1.0 | $3 \times 10^{-5}$ |
| | | 0.1 | 0.0 | $3 \times 10^{-5}$ |
| | Cosine | 0.1 | 1.0 | $3 \times 10^{-5}$ |
| | | 0.1 | 0.0 | $1 \times 10^{-4}$ |
| | Warmup-Stable-Only (WSO) | 1.0 | 1.0 | $1 \times 10^{-6}$ |
| | WSD | 1.0 | 0.0 | $1 \times 10^{-6}$ |
| | | 0.1 | 1.0 | $1 \times 10^{-4}$ |
| 8B | | 0.1 | 0.0 | $3 \times 10^{-5}$ |
| | Linear | 0.1 | 1.0 | $1 \times 10^{-5}$ |
| | | 0.1 | 0.0 | $1 \times 10^{-5}$ |
| | Cosine | 0.1 | 1.0 | $1 \times 10^{-5}$ |
| | | 0.1 | 0.0 | $1 \times 10^{-5}$ |

Table 18: SFT evaluation results after mid-training. WSO throughout pre- and mid-training generally achieves better SFT performance.

| Model | Pre-training Scheduler | $\alpha_{pre}$ | $\alpha_{mid}$ | AlpacaEval | TruthfulQA | GSM8K | DROP | AGI Eval | BBH | MMLU | Avg. |
|---|---|---|---|---|---|---|---|---|---|---|---|
| | Warmup-Stable-Only (WSO) | 1.0 | 1.0 | **79.4** | **41.8** | **29.0** | 22.7 | 21.8 | 23.1 | **35.7** | **36.2** |
| | | 1.0 | 0.0 | **79.4** | 39.9 | 27.2 | 22.0 | 21.5 | 22.7 | 35.4 | 35.4 |
| | WSD | 0.1 | 0.0 | 76.8 | 41.0 | 18.9 | 22.0 | 22.4 | **23.8** | 34.2 | 34.2 |
| 1B | | 0.1 | 1.0 | 78.7 | 40.0 | 21.2 | **23.7** | **23.1** | 23.8 | 34.4 | 35.0 |
| | Cosine | 0.1 | 0.0 | 72.9 | 38.1 | 19.9 | 17.6 | 22.1 | 17.9 | 33.9 | 31.8 |
| | | 0.1 | 1.0 | 74.3 | 37.9 | 22.2 | 17.1 | 22.6 | 19.6 | 34.0 | 32.5 |
| | Linear | 0.1 | 0.0 | 73.2 | 39.1 | 14.0 | 16.2 | 22.1 | 22.3 | 34.3 | 31.6 |
| | | 0.1 | 1.0 | 76.3 | 40.8 | 17.7 | 16.3 | 22.8 | 21.4 | 35.1 | 32.9 |
| | Warmup-Stable-Only (WSO) | 1.0 | 1.0 | 64.1 | 43.4 | **54.7** | **36.4** | **40.2** | 31.2 | 42.9 | **44.7** |
| | | 1.0 | 0.0 | 68.6 | **44.8** | 34.5 | 32.6 | 40.0 | 30.9 | 44.3 | 42.2 |
| | WSD | 0.1 | 0.0 | 66.8 | 44.1 | 40.9 | 28.3 | 36.4 | **31.5** | **49.6** | 42.5 |
| 8B | | 0.1 | 1.0 | **69.7** | 43.9 | 47.3 | 29.9 | 36.3 | 29.0 | 49.5 | 43.6 |
| | Cosine | 0.1 | 0.0 | 64.7 | 41.1 | 41.0 | 26.9 | 32.3 | 27.9 | 43.0 | 39.6 |
| | | 0.1 | 1.0 | 63.9 | 41.9 | 40.8 | 28.8 | 34.6 | 28.5 | 42.8 | 40.2 |
| | Linear | 0.1 | 0.0 | 63.9 | 42.5 | 36.8 | 28.3 | 33.6 | 29.3 | 44.9 | 39.9 |
| | | 0.1 | 1.0 | 63.8 | 41.3 | 43.5 | 30.5 | 33.0 | 31.0 | 46.8 | 41.4 |

Table 19: SFT learning rates selected for each over-trained and mid-trained 1B model based on AlpacaEval performance.

| Model | Scheduler | $\alpha_{pre}$ | $\alpha_{mid}$ | Selected SFT LR |
|---|---|---|---|---|
| | Warmup-Stable-Only (WSO) | 1.0 | 1.0 | $1 \times 10^{-5}$ |
| | | 1.0 | 0.0 | $1 \times 10^{-5}$ |
| | WSD | 0.1 | 1.0 | $3 \times 10^{-5}$ |
| 1B | | 0.1 | 0.0 | $1 \times 10^{-5}$ |
| | Linear | 0.1 | 1.0 | $1 \times 10^{-5}$ |
| | | 0.1 | 0.0 | $1 \times 10^{-5}$ |
| | Cosine | 0.1 | 1.0 | $1 \times 10^{-5}$ |
| | | 0.1 | 0.0 | $1 \times 10^{-5}$ |

Table 20: SFT evaluation results for over-trained 1B models after mid-training (pre-trained on 2T tokens, mid-trained on 500B tokens, then supervised fine-tuned).

| Model | Pre-training Scheduler | $\alpha_{pre}$ | $\alpha_{mid}$ | AlpacaEval | TruthfulQA | GSM8K | DROP | AGI Eval | BBH | MMLU | Avg. |
|---|---|---|---|---|---|---|---|---|---|---|---|
| | Warmup-Stable-Only (WSO) | 1.0 | 1.0 | 66.2 | 38.1 | **30.3** | 19.4 | **24.1** | 24.8 | **36.6** | **34.2** |
| | | 1.0 | 0.0 | 64.0 | 40.4 | 20.4 | 18.4 | 21.3 | **26.1** | 35.5 | 32.3 |
| | WSD | 0.1 | 0.0 | 64.8 | 39.8 | 15.6 | 19.5 | 21.4 | 23.7 | 36.0 | 31.5 |
| 1B | | 0.1 | 1.0 | 62.1 | 39.7 | 21.9 | 16.8 | 21.1 | 25.0 | 35.9 | 31.8 |
| | Cosine | 0.1 | 0.0 | 62.5 | 41.1 | 18.7 | **20.5** | 23.2 | 18.8 | 35.9 | 31.5 |
| | | 0.1 | 1.0 | 64.6 | **42.0** | 21.0 | 18.7 | 23.0 | 20.0 | 35.4 | 32.1 |
| | Linear | 0.1 | 0.0 | 64.7 | 39.0 | 20.2 | 19.6 | 22.6 | 24.2 | 34.8 | 32.2 |
| | | 0.1 | 1.0 | **66.8** | 39.4 | 22.3 | 19.5 | 22.6 | 23.8 | 35.0 | 32.2 |

Table 21: Mid-training configuration for 1B and 8B models.

| Hyperparameter | 1B | 8B |
|---|---|---|
| *Training Configuration* | | |
| Total training steps | 36,000 | 36,000 |
| Total tokens | 150B | 225B |
| Batch size (tokens) | 4,194,304 | 12,582,912 |
| Sequence length | 2,048 | 2,048 |

Table 22: Mid-training configurations in over-training settings for the 1B model trained on 500BT tokens. All other hyperparameters are identical to those in Table 6.

| Hyperparameter | Value |
|---|---|
| *Training Configuration* | |
| Total training steps | 30,000 |
| Total tokens | 500BT |
| Batch size (tokens) | 16,777,216 |

Table 23: Configuration for sharpness (Hessian trace) computation using Hutchinson's estimator.

| Hyperparameter | Value |
|---|---|
| Sequence length | 1,024 |
| Batch size | 1 |
| Number of views | 2 |
| Hutchinson samples | 50 |
| Maximum batches | 4,096 |
| Maximum texts | 16,192 |

