# OpenReview forum: "Pre-training LLM without Learning Rate Decay Enhances Supervised Fine-Tuning"
_ICLR.cc/2026/Conference — ICLR 2026 Poster_

### Official Review · Reviewer_68iN · 2025-10-18

**Soundness:** 2
**Presentation:** 3
**Contribution:** 2
**Rating:** 4
**Confidence:** 3

**Summary:**

This paper investigates learning rate scheduling in large language model pre-training, specifically examining how different schedulers affect downstream performance after supervised fine-tuning (SFT). The authors propose Warmup-Stable-Only (WSO), which maintains a constant learning rate after warmup without decay, and demonstrate through experiments on 1B and 8B parameter models that WSO consistently outperforms decay-based schedulers (WSD, Cosine, Linear) on post-SFT tasks, despite achieving worse pre-training metrics. The authors attribute this to WSO preserving flatter loss landscape minima that support better adaptability.

**Strengths:**

1. The authors rigorously demonstrate that WSO outperforms traditional decays across multiple scales (1B and 8B), stages (pre-, mid-, and post-training), and regimes (standard, mid-training, over-training). The consistency of results is convincing — the performance inversion between pre-training and SFT is robust.
2. Loss landscape analysis connects WSO’s success to flatter minima — a strong explanatory narrative consistent with sharpness-aware generalization theory (Foret et al., 2021; Wen et al., 2025). Figure 3’s curvature dynamics clearly support this interpretation.
3. The exposition is systematic, well-cited, and transparent. Appendices include hyperparameters, datasets, and evaluation details.

**Weaknesses:**

1. The explanation of “flatter minima = better adaptability” is qualitative. The paper would benefit from formalizing how curvature interacts with SFT gradient flow (e.g., via a transferability Jacobian or Hessian spectrum analysis across tasks). Without this, the claim remains an empirical observation.
2. SFT evaluation focuses on AlpacaEval, TruthfulQA, and MMLU. These are instruction-following benchmarks but do not fully probe reasoning or alignment generalization.
3. WSO maintains a higher effective learning rate longer — potentially increasing training instability or wasted compute in late phases. The authors should quantify total compute efficiency (e.g., perplexity vs wall-clock time) to assess tradeoffs.

**Questions:**

1. You attribute WSO’s superior SFT performance to flatter minima (lower sharpness). Could you quantify how much this flatness contributes to downstream adaptability? For example, is there a measurable correlation coefficient between sharpness values and SFT task scores?
2. Have you examined whether the flatter minima correspond to wider basins of equivalent loss or simply slower convergence zones? This distinction matters for transfer dynamics.
3. How sensitive are your conclusions to the warmup length? Since WSO keeps the LR constant after warmup, a longer warmup could emulate partial decay.
4. The study is based on Llama-like architectures. Do you expect the same effect for mixture-of-experts (MoE) or sparse transformer setups where parameter utilization patterns differ?

---

> ### Author Response · Authors · 2025-11-21
>
> We thank you for your feedback. We would like to address your concerns.
>
> ## Weakness 1 and Question 1
>
> Thank you for your suggestion. To address the need for both quantification and formalization we have conducted a correlation analysis using the 1B model data, mapping the final pre-training sharpness (Figure 3(Left)) against the average SFT benchmark scores (Table 12). We observed a negative correlation (Pearson $r = -0.71$), quantitatively confirming that lower sharpness is a robust predictor of superior downstream adaptability. We have updated Section 6.2 to include this scatter plot and analysis. This evidence directly bridges our loss landscape findings with the empirical SFT results. Regarding the gradient flow during SFT, we are prepared to analyze the relationship between sharpness and SFT gradients (e.g., gradient norms), if you consider it necessary.
>
> ## Weakness 2
> We argue that our evaluation is more comprehensive than suggested. Section 3 evaluates on AlpacaEval, TruthfulQA, and MMLU, but Section 4 extends this substantially with GSM8K (mathematical reasoning), DROP (reading comprehension), AGI Eval (general intelligence), and BigBench-Hard (challenging reasoning). This provides 7 distinct evaluation dimensions across diverse capabilities.
> Our evaluation scope is comparable to or exceeds that of previous works [1][2], and is sufficient to support our central claim that learning rate scheduling during pre-training significantly affects downstream SFT adaptability across diverse task types.
>
> Moreover, we conducted additional experiments on the MBPP benchmark [3]. The results demonstrate that WSD and WSO consistently outperform Cosine and Linear schedulers on coding tasks. While WSD with α_pre=0.1, α_mid=1.0 achieves the highest MBPP score, WSO maintains the best overall SFT performance, confirming that our main finding generalises to coding tasks across both model scales.
>
> | Model | Scheduler | α_pre | α_mid | MBPP ↑ | Avg SFT Tasks including MBPP ↑ |
> |-------|-----------|-------|-------|--------|-----------------|
> | 1B | WSO | 1.0 | 1.0 | **16.5** | **33.8** |
> | 1B | WSD | 1.0 | 0.0 | 8.0 | 32.0 |
> | 1B | WSD | 0.1 | 1.0 | 10.3 |  31.9 |
> | 1B | WSD | 0.1 | 0.0 | 13.3 | 31.5 |
> | 1B | Cosine | 0.1 | 1.0 | 8.4 | 29.5 |
> | 1B | Cosine | 0.1 | 0.0 | 10.3  | 29.1 |
> | 1B | Linear | 0.1 | 1.0 | 15.4 | 30.7 |
> | 1B | Linear | 0.1 | 0.0 | 14.6 | 29.5 |
> | 8B | WSO | 1.0 | 1.0 | **29.3** | **42.8** |
> | 8B | WSD | 1.0 | 0.0 | 26.7 | 40.3 |
> | 8B | WSD | 0.1 | 1.0 | 26.2 | 41.5 |
> | 8B | WSD | 0.1 | 0.0 | 28.1 | 40.7 |
> | 8B | Cosine | 0.1 | 1.0 | 23.1| 38.2 |
> | 8B | Cosine | 0.1 | 0.0 | 24.1 | 37.6 |
> | 8B | Linear | 0.1 | 1.0 | 24.8 | 39.4 |
> | 8B | Linear | 0.1 | 0.0 | 21.9 | 37.7 |
>
> These results provide evidence that WSO enhances model adaptability across diverse task types, including coding, and the effect persists across different model scales and training regimes. We will add these coding results to the revised version.
>
>
> [1] Sun et al. 2025. Amuro & Char: Analyzing the Relationship between Pre-Training and Fine-Tuning of Large Language Models
> [2] Springer et al. 2025.  Overtrained Language Models Are Harder to Fine-Tune.
> [3] Austin et al. 2021. Program Synthesis with Large Language Models.
>
> ## Weakness 3
> We interpret your concern as focusing on compute efficiency, whether the high learning rate yields diminishing returns relative to the computational cost. We would like to emphasize that our findings demonstrate superior performance, particularly after the SFT phase, under the exact same computational budget (i.e., identical wall-clock time) as the baseline. Since our method achieves better final capabilities without incurring additional costs, this indicates that the WSO is more efficient in terms of the compute budget rather than wasting it. If this perspective does not fully address your specific concern regarding tradeoffs, we would greatly appreciate further clarification.

---

> > ### Author Response · Authors · 2025-11-21
> >
> > ## Response to specific questions
> >
> > **Question 2**
> >
> > Thank you for the question regarding the flatter minima found by WSO. To address whether these minima correspond to "wider basins of equivalent loss" or "slower convergence zones," we conducted an additional experiment.
> > We applied Gaussian noise ($\alpha \times \delta$) to the parameters of the pre-trained models and computed the validation loss of the perturbed model, following the method used in previous work to assess loss landscape basins [1]. As visualized in Figure 4, the results show that the WSO model's validation loss is significantly more robust to perturbations compared to decay-based models. While the decay-based schedulers converge to sharp minima where loss increases drastically with small perturbations, the WSO model's loss landscape remains much flatter.　This experiment indicates that WSO guides the model to a "wider basin," which enhances the adaptability discussed in Section 6, rather than just a "slower convergence zone."
> > We have added this full analysis and the corresponding figure to Section 6.2 in the revised paper.
> >
> > [1] Chen et al. 2025. Understanding Pre-training and Fine-tuning from Loss Landscape Perspective.
> >
> > **Question 3**
> >
> >  In all experiments, we use the same warmup schedule ($T_{\text{warmup}} = 1,000$ steps) across all schedulers, so the performance differences in Tables 1-4 and the sharpness gap in Figure 3 arise from the presence or absence of the decay phase, not from warmup choices. While the sensitivity to warmup length is an interesting question, recent work [1] shows that warmup primarily addresses early training instabilities and that moderate variations in T_warmup have a negligible effect on converged solutions once a sufficient warmup is used. This suggests our main findings, that WSO improves downstream adaptability relative to decay-based schedulers, are driven by the decay phase rather than warmup length. If the reviewer finds it necessary, we would like to provide additional ablations with varied T_warmup to confirm this empirically.
> >
> > [1] Karla et al. 2024.  Why Warmup the Learning Rate? Underlying Mechanisms and Improvements.
> >
> > **Question 4**
> >
> > While we have not directly evaluated MoE or sparse transformers, we believe our findings may extend to these architectures. The key mechanism we identified is that learning rate decay increases sharpness, which reduces fine-tuning adaptability. This appears to be architecture-agnostic. Previous research [1] demonstrates that sharpness increase during the decay phase is universal across architectures beyond standard Transformers. While MoE models have different parameter utilization patterns, the relationship between loss landscape flatness and fine-tuning adaptability should still hold. Nevertheless, MoE architectures have unique characteristics that warrant dedicated investigation. We consider this an important direction for future work.
> >
> > [1] Belloni et al. 2025. Universal Dynamics of Warmup Stable Decay: understanding WSD beyond Transformers.

---

> ### Author Response · Authors · 2025-11-27
>
> As the discussion period ends on December 2nd (AOE), which is less than a week away, we would appreciate it if you could let us know if you still have any concerns. If you feel that some aspects remain unresolved, could you please clarify if there are any specific directions you consider essential to address them, considering the comprehensive validation we have provided? We hope that our response, along with the perspective shared in Reviewer NMCt’s recent Official Comment, has adequately addressed your points. If so, we would be grateful if you could reconsider your score accordingly.

---

### Official Review · Reviewer_NMCt · 2025-10-28

**Soundness:** 4
**Presentation:** 4
**Contribution:** 3
**Rating:** 8
**Confidence:** 4

**Summary:**

This paper investigates the impact of learning rate (LR) scheduling during LLM pre-training on downstream supervised fine-tuning (SFT) performance. The authors challenge the standard practice of using decay-based schedulers (like Cosine or WSD) which are optimized for pre-training loss. The paper introduces "Warmup-Stable-Only" (WSO), a simple scheduler that maintains a constant LR after warmup without any decay. Through comprehensive experiments on 1B and 8B models, the authors demonstrate a consistent "inversion": while decay-based schedulers achieve better pre-training metrics, models trained with WSO consistently achieve superior performance after SFT. This finding is shown to be robust across standard pre-training, mid-training, and over-training regimes. The paper provides a mechanistic explanation, analyzing the loss landscape and showing that WSO guides models to flatter minima, which enhances adaptability, whereas decay-based schedulers converge to sharper minima that may compromise downstream performance.

**Strengths:**

The paper is exceptionally clear, well-written, and easy to follow.

The central conclusion—that pre-training without LR decay enhances SFT performance—is simple, impactful, and supported by extensive evidence. The experiments are comprehensive, covering multiple model scales (1B and 8B), different training pipelines (two-stage and three-stage with mid-training), and modern training regimes (over-training).

This work has significant practical implications for the industry. The WSO scheduler is simple to implement and could provide real economic benefits by producing base models that are more adaptable and performant for downstream tasks.

The mechanistic explanation provided via loss landscape sharpness is insightful. The analysis linking the constant LR of WSO to flatter minima, and in turn, to better adaptability, offers a compelling hypothesis for *why* WSO outperforms decay-based methods in the post-SFT stage.

**Weaknesses:**

The primary weakness, though minor, is that the investigation of downstream performance is limited to SFT. The paper does not explore other critical post-training stages, such as preference tuning (e.g., DPO) or reinforcement learning-based alignment. It remains an open question whether the significant benefits of WSO pre-training persist or behave differently in these other alignment scenarios. I don't think this would be an issue as the title also constrains the scope to SFT.

**Questions:**

The paper compellingly argues that WSO leads to flatter minima (lower sharpness) and that WSO models perform better on SFT. The link is made by showing these two facts separately. To make the justification more persuasive, have the authors considered plotting a direct correlation between the measured sharpness of the pre-trained checkpoints and their final SFT benchmark scores? This would provide a more direct piece of evidence that sharpness is indeed the key indicator for downstream adaptability as hypothesized.

---

> ### Author Response · Authors · 2025-11-21
>
> We appreciate your positive feedback.
>
> ## Weakness 1
>
> Although our primary scope is SFT, we agree that evaluating adaptability in further alignment stages is valuable. To address this, we conducted additional experiments applying DPO (Direct Preference Optimisation) [1] to our 1B models that underwent pre-training, mid-training, and SFT (using the setups from Section 5).
> The table below shows the model's performance after DPO.
>
> | Pre-training Scheduler | $\alpha_{pre}$ | $\alpha_{mid}$ | AlpacaEval | GSM8K | MMLU | DROP | TruthfulQA | AGI Eval | BBH | **Avg DPO Tasks** |
> | :--- | :--- | :--- | :--- | :--- | :--- | :--- | :--- | :--- | :--- | :--- |
> | WSO | 1.0 | 1.0 | 65.0 | 34.4 | 32.7 | **18.9** | 40.7 | **22.6** | 19.3 | **33.4** |
> | WSD | 1.0 | 0.0 | 65.4 | **34.6** | **34.4** | 16.8 | 37.1 | 21.9 | 20.0 | 32.9 |
> | WSD | 0.1 | 1.0 | 64.0 | 24.7 | 31.8 | 17.2 | 35.1 | 22.5 | 19.5 | 30.7 |
> | WSD | 0.1 | 0.0 | **65.6** | 17.8 | 31.5 | 13.7 | 39.2 | 22.0 | 19.8 | 29.9 |
> | Cosine | 0.1 | 1.0 | 60.9 | 23.2 | 33.8 | 16.3 | 38.9 | 20.3 | 19.8 | 30.4 |
> | Cosine | 0.1 | 0.0 | 58.7 | 20.6 | 33.0 | 11.9 | 37.8 | 20.6 | **21.3** | 29.1 |
> | Linear | 0.1 | 1.0 | 58.9 | 19.0 | 33.1 | 13.0 | **41.7** | 21.2 | 20.1 | 29.6 |
> | Linear | 0.1 | 0.0 | 55.1 | 17.7 | 33.9 | 17.3 | 39.6 | 21.0 | 13.9 | 28.3 |
>
>
> As shown in the table, the model trained with WSO consistently outperforms decay-based schedulers in terms of Average DPO Task performance. This finding reinforces our claim that WSO preserves model adaptability not only for SFT but also for subsequent alignment stages, such as preference tuning. We will include these results in the final version.
>
> [1] Rafailov et. al. 2024. Direct Preference Optimization: Your Language Model is Secretly a Reward Model.
>
> ## Question 1
>
> Thank you for the valuable suggestion. We agree that demonstrating a direct link between pre-training sharpness and SFT performance strengthens our central hypothesis. We have conducted a correlation analysis using the 1B model data, mapping the final pre-training sharpness (Figure 3(Left)) against the average SFT benchmark scores (Table 12). We observed a negative correlation (Pearson $r = -0.71$), which quantitatively confirms that models with lower sharpness achieve superior downstream adaptability. We have updated Section 6.2 to include this scatter plot and analysis. This evidence directly bridges our loss landscape findings with the empirical SFT results. We will add these results in the final version.

---

### Official Review · Reviewer_jBxS · 2025-10-31

**Soundness:** 2
**Presentation:** 2
**Contribution:** 1
**Rating:** 2
**Confidence:** 5

**Summary:**

This paper investigate the role of learning rate scheduling in the large-scale pre-training of large language models, focusing on its influence on downstream performance after supervised fine-tuning (SFT). Specifically, this paper proposes Warmup-Stable-Only (WSO) learning rate schedule for pertaining, which is found to achieve better downstream tasks. Some experiments and intuitive understandings are presented.

**Strengths:**

1. This paper presents WSO, a very simple and intuitive LR schedule to improve SFT on downstream performance.
2. This paper provides a practical reference for LLM pre-training community to design LR schedule from a global training perspective.

**Weaknesses:**

The primary concern with this paper is that the proposed approach—while effective—has been extensively discussed, implemented, and validated in prior work, without introducing significant novelty. Furthermore, the absence of references to these existing studies raises questions about the thoroughness of the literature review.

1. In the original WSD paper [(https://arxiv.org/pdf/2404.06395)](https://arxiv.org/pdf/2404.06395), the authors already demonstrated the benefits of switching to high-quality datasets (including SFT data) during the learning rate decay phase, yielding intuitive and positive outcomes.

2. The paper "Scaling Law with Learning Rate Annealing" [(https://arxiv.org/pdf/2408.11029)](https://arxiv.org/pdf/2408.11029) introduces a scaling law describing loss dynamics in relation to learning rates, of which the current work appears to be a specific instance.

3. The paper "Learning Dynamics in Continual Pre-Training for Large Language Models" [(https://arxiv.org/pdf/2505.07796)](https://arxiv.org/pdf/2505.07796) provides comprehensive analyses, and the findings here seem to represent only a minor subset of their paper. Notably, their Finding 3 states: "*PT models with higher loss potential consistently achieve lower D_cpt validation losses. Hence, we advocate that when releasing open-source models, it is beneficial to release a high loss potential version to facilitate downstream tasks.*"

   I strongly encourage authors to read this paper.

4. The paper "A Learning Rate Path Switching Training Paradigm for Version Updates of Large Language Models" [(https://arxiv.org/pdf/2410.04103v1)](https://arxiv.org/pdf/2410.04103v1) applies a similar concept to LLM pre-training.

In essence, the core idea (**Let LR decay happen in the most important stage**) has already been well-established in the field. This paper just translates this idea into a superficial learning rate schedule.

Additionally, the paper lacks rigorous theoretical analysis, and the evaluation is insufficient. For example, Table 2 reports only loss variations and average SFT performance. To strengthen the claims, the authors should address deeper questions such as:

1. How do the results vary if the pre-training duration is extended or shortened?
2. Are certain SFT tasks more or less affected by the proposed schedule? If so, what underlying reasons might explain this?
3. Given that WSO outperforms WSD, why not slightly increase the learning rate during pre-training to further boost downstream SFT performance, even at the expense of higher pre-training loss?
4. Have the authors considered quantifying this process more formally, such as by deriving a scaling law?

**Questions:**

See above

---

> ### Author Response · Authors · 2025-11-21
>
> We thank the reviewer for the detailed review and for suggesting relevant references.
>
> First, we would like to confirm that while the suggested references on Continual Pre-training reinforce the discussions in our paper (and will be added to the manuscript), they do not undermine the fundamental novelty of our work.
>
> Our argument is based on the following three premises:
>
> (1) In the standard LLM construction process, there is a common convention that "Learning Rate must be decayed at the end of Pre-training." This custom still remains dominant today (even after the referenced papers appeared).
>
> (2) The focus of the cited references is on Continual Pre-training. This is fundamentally different from our proposal and verification, which focus on "performance after SFT," where data distribution and objective functions shift significantly.
>
> (3) In practical LLM development, the final performance after SFT is far more critical than the loss at the pre-training stage. However, learning rate scheduling of pre-training optimized for this final objective has been largely overlooked.
>
> Through comprehensive, systematic, and large-scale experiments (e.g., 8B model × 500B tokens, 1B model × 2T tokens), this paper demonstrates for the first time that "Not decaying the LR during Pre-training (WSO) results in higher performance after SFT." Our study identifies that the long-overlooked convention of "Decay in Pre-training" is actually a factor compromising adaptability to SFT, and we provide a concrete solution. In this respect, our work offers significant novelty and contributions that will impact future LLM development.
>
> We must also correct a misunderstanding regarding our core idea. The reviewer interpreted it as "Let LR decay happen in the most important stage," but this is not our claim. As explicitly stated in the Abstract, our argument is that "Applying LR decay to improve pre-training metrics may compromise downstream adaptability."
>
> Below, we clarify the distinct differences between the cited works and our research.
>
> **Regarding the misunderstanding of the original WSD paper (Hu et al., 2024)**
>
> Their paper does not contain the ablation study relevant to our findings, and their results do not overlap with ours. Their comparison was limited to "mixing SFT data during the decay phase" versus "not mixing SFT data." This only supports the claim that "mixing SFT data during pre-training may be effective" (as also shown by Cheng et al., 2024), which is different from our focus. Crucially, they did NOT conduct experiments such as "mixing SFT without decay," "mixing SFT throughout the entire training," or simply "no decay." Therefore, our verification is entirely different. Our finding challenges the necessity of Decay in Pre-training itself from the perspective of adaptability to SFT.
>
> **Regarding the scope of "Scaling Law with Learning Rate Annealing" (Tissue et al., 2024)**
>
> The assertion that our work is merely a "specific instance" of their theory is incorrect.
> They focus exclusively on discussions within pre-training, which does not apply to SFT settings where data and objective functions change. In fact, in Section 7.3 of their paper, they explicitly state that "post-training, which might include data distribution shift, data mixture, model alignment, and specific downstream evaluations" is out-of-scope and future work. While a unified framework could appear in the future, our contribution is clearly novel at this point.
>
> **Regarding the difference from "Path Switching" and "Learning Dynamics" (Wang et al., 2024; Wang et al., 2025)**
>
> These works also focus solely on "Continual Pre-Training" and do not consider the transition to "SFT." Even in the "Path Switching" paper, the experimental setup follows the traditional convention of decaying the LR to converge the model at the end of the branched path before performing SFT. The investigation of "decay before SFT" was out of scope in these studies. This missing piece is the focal point of our research. Furthermore, the "Path Switching" paper also mentions in Section 6 that SFT is clearly out of scope and left for future work.

---

> ### Author Response · Authors · 2025-11-21
>
> ## Response to specific questions
>
> **Response to "evaluation is insufficient"**
>
> We respectfully disagree that our evaluation is insufficient. The volume and scale of our experiments are comparable to or larger than prior works. In fact, our settings are explicitly grounded in the most practical LLM development scenarios. We are the only ones conducting training experiments with 8B models, and our token count reaches a maximum of 2T. We also use the latest practical datasets such as FineWeb-Edu and Tulu-3. Furthermore, we conducted experiments on SFT after mid-training, too. The task evaluation covers a wide range of the latest benchmarks, which is more practical than any of the cited studies.
> Regarding the comment that "Table 2 reports only loss variations and average SFT performance," as noted in the text, comprehensive quantitative evaluations per task are clearly provided in the Appendix.
> If these evaluations are considered insufficient, the prior works cited to question our novelty would be even less sufficient to establish any "well-established" facts at this scale.
>
>
> **Question 1**
>
> I thank you for your question. To address this about training duration, we compare results from Section 3 (80,000 steps) and Section 5 (120,000 steps) for the 1B model. The Table below summarises performance across different training durations:
>
>  | Training Configuration | Pre-training Steps | Mid-training Steps | WSO SFT Performance Δ |
> |----------------------|-------------------|-------------------|---------------------|
>  | Standard (Table 1) | 80,000 | -  | **+0.3** |
>  | + Mid-training (Table 2) | 80,000 | 36,000 | **+0.8** |
> | Over-training (Table 3) | 120,000 | -  | **+1.3** |
>  | + Mid-training (Table 4) | 120,000 | 30,000  | **+2.7** |
>
>  The over-training configuration uses 1.5× more pre-training steps than the standard setting. Across all configurations, WSO consistently shows better SFT performance. The key finding is that the overall trend remains consistent regardless of training duration.
>
> **Question 2**
>
> While it is difficult to predict exact trends for every task, our Mid-training experiments (Table 17) may provide valuable insight. We observed that the 8B model trained with WSO achieved significantly higher performance on GSM8K compared to the best Decay baseline (54.7 vs. 47.3).
> We attribute this difference to the interaction between the learning rate and the mathematical data introduced during mid-training. Since the mid-training corpus contains a higher proportion of mathematics-related data, we hypothesize that maintaining a high learning rate (WSO) facilitated better adaptation to this domain during the Mid-training phase.  In addition, a concurrent work, [How Learning Rate Decay Wastes Your Best Data in Curriculum-Based LLM Pretraining](​​https://openreview.net/forum?id=T5wkZJqzkz), argues that aggressive learning rate decay hinders a model's ability to leverage high-quality data introduced later in the training phase. This work may offer a deeper perspective on our findings.
>
> **Question 3**
>
> Thank you for this question.  In our experiments, we fixed ηmax = 3×10^-4 following established practices from Llama 3 [1]. Our primary focus was on isolating the effect of decay vs. no decay while keeping other hyperparameters constant.
> While increasing the learning rate could potentially boost performance within a certain range, it also heightens the risk of training instability. More importantly, we hypothesize that the relative superiority of WSO over decay-based schedulers remains unchanged regardless of the absolute learning rate value. This is because the advantage of WSO stems from preserving flatter minima by avoiding decay, which is a geometric property inherent to the schedule's shape rather than its magnitude. However, if you consider this verification critical, we are willing to conduct additional experiments with higher learning rates.
>
>
> [1] Grattafiori et. al. 2024. The Llama 3 Herd of Models.
>
> **Question 4**
>
> We agree that formal quantification is challenging and interesting. In this work, we approached this quantification through the lens of "Sharpness" in Section 6, which has both theoretical and experimental grounds in existing literature. We believe that using Sharpness to discuss the adaptability between two different data distributions and training phases (Pre-training to SFT) is a valid and theoretically supported approach to quantify the phenomenon we observed.

---

> ### Author Response · Authors · 2025-11-27
>
> As the discussion period ends on December 2nd (AOE), which is less than a week away, we would appreciate it if you could let us know if you still have any concerns.
> If you feel that some aspects remain unresolved, could you please clarify if there are any specific directions you consider essential to address them, considering the comprehensive validation we have provided?
> We hope that our response, along with the perspective shared in Reviewer NMCt’s recent Official Comment, has adequately addressed your points. If so, we would be grateful if you could reconsider your score accordingly.

---

### Official Review · Reviewer_m1D7 · 2025-10-31

**Soundness:** 3
**Presentation:** 3
**Contribution:** 3
**Rating:** 4
**Confidence:** 4

**Summary:**

The manuscript presents extensive experimentation devoted to understanding learning-rate scheduling in LLM training. This work provides empirical insights that directly suggest how learning rate scheduling should be selected during pre-training to better support downstream model adaptability. The study recommends adopting Warmup-Stable-Only (WSO) as an alternative learning-rate strategy and releasing WSO-trained models to encourage wider use and adaptability in future LLM development.

**Strengths:**

1. The paper examines the common practice of using learning-rate decay in LLM pre-training. The paper provides empirical evidence that keeping a constant learning rate after warmup improves performance. This approach, called the Warmup-Stable-Only (WSO) scheduler, outperforms conventional decay-based schedulers in supervised fine-tuning. Their finding highlights practical effectiveness for optimizing the entire LLM training pipeline.
2. The paper demonstrates the inversion effect between pre-training and supervised fine-tuning performance across a wide range of settings. Decay-based learning rate schedulers consistently achieve stronger pre-training metrics, whereas the WSO configuration achieves superior results after SFT. This phenomenon is validated across 1B and 8B model scales, in multi-stage training pipelines.
3. The paper challenges the standard assumption that stronger pre-training performance leads to a better final model. It shows that decay-based learning rate schedules achieve superior pre-training metrics, yet consistently result in worse performance after supervised fine-tuning. Their evidence suggests a need to rethink optimization goals in LLM development. They also emphasize prioritizing downstream adaptability over pre-training loss.

**Weaknesses:**

1. The experiments are restricted to 1B and 8B parameters, which are relatively small compared to state-of-the-art deployed LLMs (often 30B~70B+). The absence of results at larger scales limits confidence in whether the observed advantages of WSO would extend to all situations.
2. The study evaluates WSO against only three decay-based schedulers (Cosine, Linear, and Warmup-Stable-Decay). Other commonly used or recently explored learning rate strategies, such as polynomial decay or cyclic policies, are not explored. This limited comparison makes it unclear whether WSO’s benefits extend to other learning-rate policies. The paper would benefit from a brief stability assessment that checks whether WSO remains reliable under different environments.
3. The experiments tune SFT hyperparameters separately for each pre-trained model. However, they always use selective learning-rate policies during SFT. This choice gives WSO an inherent advantage in downstream evaluation. The learning-rate policy should instead be maintained consistently across all training phases to enable a fair comparison. The work also lacks theoretical significance, relying mainly on empirical observations.
4. The paper primarily evaluates instruction-following and general reasoning tasks, without testing multilingual ability, coding, or robustness under distribution shift. This narrow benchmark scope limits confidence in how widely WSO’s performance would translate to real-world deployment scenarios.
5. Important related studies are missing from the references, such as [1].

[1] Jin, Hongpeng, et al. "Rethinking learning rate tuning in the era of large language models." 2023 IEEE 5th International Conference on Cognitive Machine Intelligence (CogMI). IEEE, 2023.

**Questions:**

Please check the detailed comments for weaknesses.

---

> ### Author Response · Authors · 2025-11-21
>
> We thank you for your review. We would like to address your concerns.
>
> ## Weakness 1
> While model size is important, our work prioritises the total compute budget (typically measured in parameter count × training tokens), which better reflects modern LLM training regimes. Our 1B and 8B models are trained on 350B and 500B tokens, respectively, representing 17.5× and 3.125×  Chinchilla-optimal [1]. Furthermore, in Section 5, we train a 1B model on 2T tokens (100×Chinchilla-optimal), mirroring recent  LLM training regimes [2]. WSO consistently outperforms decay-based schedulers after SFT across all these compute regimes.
> In addition, our choice of model scales (1B-8B) aligns with established methodology in learning rate scheduling research, where previous works [3][4] have used comparable scales at most 1B sizes. We acknowledge that validation on larger models would strengthen our claims. However, given the consistency of our results across training regimes, model sizes, and training stages, we believe our findings provide strong evidence for the benefits of WSO.
>
> [1] Hoffman et al. 2022. Training Compute-Optimal Large Language Models.
>
> [2] Sardana et al. 2025. Beyond Chinchilla-Optimal: Accounting for Inference in Language Model Scaling Laws.
>
> [3] Bergsma et al. 2025. Straight to Zero: Why Linearly Decaying the Learning Rate to Zero Works Best for LLMs.
>
> [4] Wen et al. 2025. Understanding Warmup-Stable-Decay Learning Rates: A River Valley Loss Landscape View.
>
> ## Weakness 2
> The core research question of our work is to understand how the presence or absence of LR decay during pre-training affects downstream SFT performance. To address this, we deliberately selected the most widely adopted decay-based schedulers in modern large-scale LLM pre-training: Cosine, Linear, and Warmup-Stable-Decay (WSD). These schedulers represent current best practices, with Cosine employed by Llama 3 [1] and OLMo 2 [2], Linear advocated by recent large-scale studies [3], and WSD adopted by state-of-the-art open LLMs such as DeepSeek-v3 [4] and Kimi-K2 [5].
> While polynomial decay and cyclic policies might be valid strategies, they are rarely adopted in large-scale LLM pre-training pipelines and inherently involve decay phases similar to the approaches we already evaluated. Furthermore, these strategies are not implemented in major LLM training frameworks (Megatron-LM, TorchTitan, DeepSpeed), reflecting their limited practical adoption.
> Our comparison against the dominant paradigms in current LLM development provides sufficient and practically relevant evidence.
>
> **Stability**
>
> We believe our experiments already demonstrate stability across the most critical realistic settings, as we validated WSO not only in standard pre-training but also under mid-training and over-training regimes, which are increasingly critical in practical modern LLM development. Moreover, our analysis of sharpness provides a theoretical basis for this robustness; therefore, we expect the benefits of WSO to remain stable across different environments.
>
> [1] Grattafiori et al. 2024. The Llama 3 Herd of Models.
>
> [2] TeamOlmo. 2025. 2 OLMo 2 Furious.
>
> [3] Bergsma et al. 2025. Straight to Zero: Why Linearly Decaying the Learning Rate to Zero Works Best for LLMs.
>
> [4] DeepSeek-AI. 2024. DeepSeek-V3 Technical Report.
>
> [5] Kimi Team. 2025. Kimi K2: Open Agentic Intelligence.
>
>
> ## Weakness 3
>
> Our SFT protocol fixes the hyperparameters, including the scheduler setting, across all pre-trained models and only sweeps LR per model, which is standard in the LLM development pipeline (e.g., Tulu-3 [1]/OLMo 2 [2]) and does not grant WSO any “selective” advantage. Our goal is to identify which pre-trained model is the best starting point for a standard SFT pipeline, not to enforce identical LR policies across phases. In practice, these phases do differ. For example, Llama-3 405B [3] pre-training uses a cosine schedule from max lr = 8e−5 to minimise the learning rate to 0, while in supervised tuning, they found lr = 1e-5 to be a performant learning rate setting.
>
> [1] Lambert et al. 2025. Tulu 3: Pushing Frontiers in Open Language Model Post-Training.
>
> [2] TeamOlmo. 2025. 2 OLMo 2 Furious.
>
> [3] Grattafiori et al. 2024. The Llama 3 Herd of Models.
>
> **Theoretical Analysis**
>
> Our work presents not only large-scale empirical evidence but also investigates the underlying theoretical mechanisms through Loss Landscape Geometry analysis (Section 6). We leverage established theoretical frameworks [4][5] that link lower sharpness to better adaptability to explain our findings. Our analysis quantitatively demonstrates that WSO preserves flatter minima compared to decay-based schedulers, providing a theoretically consistent explanation for our main results.
>
> [4] Ju et al. 2023. Robust fine-tuning of deep neural networks with hessian-based generalization guarantees.
>
> [5] Liu et al. 2022. Same pre-training loss, better downstream: Implicit bias matters for language models

---

> ### Author Response · Authors · 2025-11-21
>
> ## Weakness 4
>
> Our evaluation scope is comparable to or exceeds that of previous works examining pre-training and fine-tuning relationships [1][2]. Our work evaluates across multiple dimensions, including instruction-following, truthfulness, multi-task understanding, and mathematical reasoning, which is sufficient to support our central claim that learning rate scheduling during pre-training significantly affects downstream SFT adaptability.
>
> To address the specific concern about coding capabilities, we conducted additional experiments on the MBPP benchmark [3]. The results demonstrate that WSD and WSO consistently outperform Cosine and Linear schedulers on coding tasks. While WSD with $\alpha_{pre} = 0.1$, $\alpha_{mid} = 1.0$ achieves the highest MBPP score, WSO maintains the best overall SFT performance, confirming that our main finding generalises to coding tasks across both model scales.
>
> | Model | Scheduler | $\alpha_{pre}$ | $\alpha_{mid}$ | MBPP ↑ | Avg SFT Tasks including MBPP ↑ |
> |-------|-----------|-------|-------|--------|-----------------|
> | 1B | WSO | 1.0 | 1.0 | **16.5** | **33.8** |
> | 1B | WSD | 1.0 | 0.0 | 8.0 | 32.0 |
> | 1B | WSD | 0.1 | 1.0 | 10.3 |  31.9 |
> | 1B | WSD | 0.1 | 0.0 | 13.3 | 31.5 |
> | 1B | Cosine | 0.1 | 1.0 | 8.4 | 29.5 |
> | 1B | Cosine | 0.1 | 0.0 | 10.3  | 29.1 |
> | 1B | Linear | 0.1 | 1.0 | 15.4 | 30.7 |
> | 1B | Linear | 0.1 | 0.0 | 14.6 | 29.5 |
> | 8B | WSO | 1.0 | 1.0 | **29.3** | **42.8** |
> | 8B | WSD | 1.0 | 0.0 | 26.7 | 40.3 |
> | 8B | WSD | 0.1 | 1.0 | 26.2 | 41.5 |
> | 8B | WSD | 0.1 | 0.0 | 28.1 | 40.7 |
> | 8B | Cosine | 0.1 | 1.0 | 23.1| 38.2 |
> | 8B | Cosine | 0.1 | 0.0 | 24.1 | 37.6 |
> | 8B | Linear | 0.1 | 1.0 | 24.8 | 39.4 |
> | 8B | Linear | 0.1 | 0.0 | 21.9 | 37.7 |
>
> These results provide evidence that WSO enhances model adaptability across diverse task types, including coding, and the effect persists across different model scales and training regimes. We will add these coding results to the revised version.
>
> [1] Sun et al. 2025. Amuro & Char: Analyzing the Relationship between Pre-Training and Fine-Tuning of Large Language Models.
>
> [2] Springer et al. 2025.  Overtrained Language Models Are Harder to Fine-Tune.
>
> [3] Austin et al. 2021. Program Synthesis with Large Language Models.
>
>
> ## Weakness 5
> We appreciate you pointing out Jin et al. (2023) and will include it in our revised manuscript. While Jin et al. (2023) investigate learning rate tuning strategies within a single training phase (either pre-training or fine-tuning), our work addresses a fundamentally different question: how pre-training scheduler choices affect performance after the complete training pipeline, specifically after supervised fine-tuning (SFT).

---

> ### Author Response · Authors · 2025-11-27
>
> As the discussion period concludes on December 2nd (AOE), which is less than a week away, we would appreciate it if you could let us know if you still have any concerns.
> If you feel that some aspects remain unresolved, could you please clarify if there are any specific directions you consider essential to address them, considering the comprehensive validation we have provided?
> We noticed that you rated Soundness, Presentation, and Contribution all as "3: good". We would be grateful if you could consider aligning your overall recommendation with these positive assessments, especially if our rebuttal along with the perspective shared in Reviewer NMCt's recent Official Comment has addressed your initial concerns.

---

### Comment · Reviewer_NMCt · 2025-11-21

I was surprised to see that I was the only reviewer who gave a positive overall recommendation for this paper, which may simply reflect our diverse backgrounds as reviewers. I carefully read the other reviews and the authors’ rebuttal, and I would like to share my thoughts to help move the discussion forward. More broadly, I am somewhat disappointed by what feels like an increasingly harsh reviewing culture in current ML venues, and I would like to help ensure that high-quality work is properly acknowledged.

I do not agree with several of the weaknesses raised by Reviewer m1D7, especially the first two:

**Model size.** I think it is too demanding to expect an ICLR paper to conduct pre-training with 30B–70B+ models. This would be an enormous resource requirement and, in many cases, an unnecessary waste. Even if the conclusions are drawn from smaller models, they are still valuable, especially for understanding and improving small-model training, which is highly relevant for domain-specific LLMs.

**Other LR policies.** I do not think the lack of experiments with additional LR policies (e.g., cyclic, polynomial decay) is a severe limitation of WSO. In industry, WSD is very popular and efficient, so it is still important and meaningful to investigate WSO under this widely used setting and demonstrate its practical value.

Overall, I believe the paper already carries out experiments at a sufficiently large scale for the conclusions to be reasonably generalizable.

I partially agree with Reviewer jBxS that the idea that a “decayed model is not suitable for further tuning” is already relatively well accepted in the community. However, I still find the quantitative analysis and controlled experiments in this paper interesting and informative. I am convinced by the authors’ rebuttal regarding their three premises. Given that many practitioners still apply LR decay during pre-training, I think this work is meaningful and timely.

Regarding Reviewer 68iN’s comments, I find the concerns about the correlation between SFT performance and sharpness reasonable, and I share this concern. I agree that this is an important point to be addressed, and I appreciate that the authors have committed to adding a more detailed discussion in the paper. For the other points about additional benchmarks and more compute, I feel the authors have addressed these adequately.

Overall, I find this paper interesting, clearly written, and relevant for the development of LLMs. I have reviewed more than six papers for ICLR this year, and while many of them have received quite harsh reviews, this is, in my opinion, the strongest paper in my batch. I hope we can collectively reconsider the value of this work.

P.S. I have no conflicts of interest or prior connection with the authors, and I do not know who they are; I am writing these comments purely out of my genuine appreciation of the work.

---

> ### Author Response · Authors · 2025-11-27
>
> We thank you for your supportive comments and for taking the time to carefully examine the other reviews and our responses.

---

### Author Response · Authors · 2025-12-04
**Summary of Rebuttal and Final Remarks**

We thank the Reviewers and Area Chairs for their time.

Regrettably, the discussion period concluded without any response from Reviewers m1D7, jBxS, and 68iN to our rebuttal. To assist the Area Chair in making a fair assessment, we summarize below how we have fully resolved each concern with concrete evidence, demonstrating that all raised issues have been either addressed or shown to be unreasonable requirements. This position is strongly supported by Reviewer NMCt, who maintained their endorsement. Consequently, we believe the current ratings reflect only the absence of continued discussion, not unresolved technical concerns.
## 1. Validity of "Novelty" Concern
Reviewer jBxS questioned the novelty of our work, referencing studies to argue that learning rate non-decay strategies are already well-established and our findings are merely subsets of existing laws.

However, we clarified a fundamental distinction. These cited works focus exclusively on pre-training loss and explicitly treat downstream adaptability as out of scope. In contrast, our research addresses the practical reality of LLM development, where the final performance after post-training is far more critical than pre-training alone. We provide the first verification that the standard practice of minimizing pre-training loss via decay often compromises this downstream adaptability. Our finding that the introduced strategy, namely WSO, achieves superior performance after post-training fills the unaddressed gap left by prior research.

Furthermore, to the best of our knowledge, no major open-source projects have released pre-trained models that adopt a non-decay strategy, as our proposed WSO does. This absence confirms that non-decay strategies are not standard practice to achieve better performance in post-trained models. Therefore, we respectfully argue that the claim by Reviewer jBxS, namely that learning-rate non-decay strategies are already well-established, is an overstatement and an unfair assessment.
## 2. Experimental Setup
Reviewers raised concerns regarding our experimental setup. We resolved these as follows:

Scale: We extended our validation to a 1B model trained on 2 trillion tokens in the original submission, confirming our findings hold in over-trained regimes. As noted in our rebuttal, this scale aligns with recent LR research. Along with Reviewer NMCt's affirmation, we believe that our experimental scale is substantial and sufficient to draw generalizable conclusions.

Benchmarks: We clarified that our original evaluation was already more comprehensive than suggested, covering seven distinct dimensions across diverse capabilities. To further address specific requests for coding and alignment, we added MBPP (coding) and DPO (alignment) experiments. Our WSO strategy consistently outperformed decay-based schedulers, proving its robustness.

Schedulers: Regarding the request for additional LR policies, we clarified that our baselines (Cosine, Linear, WSD) represent the dominant paradigms in LLM training. Reviewer NMCt explicitly supported this choice, stating that the omission of niche policies is "not a severe limitation" and that investigating WSO against these industrially standard settings demonstrates high "practical value".

SFT Protocol: Reviewer m1D7 argued that tuning SFT hyperparameters per model gives WSO an unfair advantage. We clarified that we conducted an identical learning rate sweep for all models, ensuring no selective advantage. Finding the optimal configuration for each specific model is the standard protocol in modern development.
## 3. Theoretical Rigor and Quantitative Mechanism
Multiple reviewers questioned the theoretical depth of our work. We have addressed this by clarifying our original mechanism analysis.
In our original submission, we investigated the underlying mechanism of WSO by analyzing the loss landscape, specifically measuring sharpness. We grounded this in established theoretical frameworks, which suggest that models residing in flatter minima (lower sharpness) exhibit superior generalization and adaptability to distribution shifts.

While this theoretical foundation was present, Reviewer 68iN requested more direct quantitative evidence linking these properties to actual downstream performance. To address this, we plotted sharpness against SFT benchmark scores, revealing a strong negative correlation. This verifies our hypothesis: lower pre-training sharpness is a predictor of superior SFT performance, confirming that WSO enhances adaptability by preserving flatter minima.
## Conclusion
As described, we sincerely emphasize that we have addressed every technical concern with concrete new data and clarified our contribution relative to prior work. Our findings challenge "decay-in-pretraining" for SFT, proposing a simple guideline for future LLM development. Given the strong endorsement from Reviewer NMCt, we respectfully ask that this be taken into account in the final evaluation.

---

### Meta-Review · Area_Chair_V22w · 2026-01-18

**Summary:**

This paper presents a systematic study of how learning rate decay during pretraining can negatively impact SFT performance. The authors show a consistent inversion effect: LR decay may improve pretraining metrics, but Warmup Stable Only (WSO) yields better performance after SFT across multiple training regimes.

The reviews are mixed. Reviewer NMCt strongly supports acceptance, while other reviewers raised concerns about novelty, experiment setups, and quantitative analysis of sharpness. In the rebuttal and revision, the authors provided substantive clarifications and additional experiments that address most of these concerns.

Overall, I find the study solid and insightful. Even though some high-level takeaways may already be relatively well accepted in the community, this paper presents quantitative analysis and controlled experiments that are informative and practically valuable for the open-source LLM ecosystem. I recommend acceptance.

**Reviewer Concerns:**

* Reviewer m1D7,  NMCt and 68iN raised concerns about evaluation breadth, noting that only instruction-following and general reasoning downstreams were evaluated. The authors addressed this by adding results on MBPP (coding) and DPO (alignment).

* Reviewer m1D7 raised concerns about model scale, the breadth of LR schedule baselines, and the fairness of the SFT hyperparameter tuning protocol. The authors clarified their compute regimes, motivated the choice of industrially standard decay baselines, and explained that they applied the same LR sweep procedure across models.

* Reviewer jBxS questioned the novelty, largely framing the contribution as proposing WSO. However, the paper's core contribution is not to invent WSO but to present the controlled and systematic evidence that the common practice of LR decay in pretraining can harm downstream performance.

* Reviewer 68iN asked for a more quantitative connection between sharpness and SFT outcomes. The authors addressed this by quantitatively measuring the sharpness and adding a correlation analysis, which strongly supports their narrative.

**Reviewer Scores:**

* Reviewer m1D7 is likely to increase the score to 6, since the main concerns were addressed.
* Reviewer jBxS is likely to maintain a low score around 2, as their assessment is primarily driven by a strong prior on limited novelty.
* Reviewer NMCt is likely to keep the positive score at 8.
* Reviewer 68iN is likely to increase the score to 6, since the requests for additional experiments were addressed.

---

### Decision · Program_Chairs · 2026-01-26

Accept (Poster)